# Proline–Proline Dyad in the Fusion Peptide of the Murine β–Coronavirus Spike Protein’s S2 Domain Modulates Its Neuroglial Tropism

**DOI:** 10.3390/v15010215

**Published:** 2023-01-12

**Authors:** Abass Alao Safiriyu, Vaishali Mulchandani, Mohammed Nahaf Anakkacheri, Debnath Pal, Jayasri Das Sarma

**Affiliations:** 1Department of Biological Sciences, Indian Institute of Science Education and Research Kolkata, Mohanpur 741246, India; 2Department of Computational and Data Sciences, Indian Institute of Science, Bengaluru 560012, India

**Keywords:** β-coronavirus, recombinant strain mouse hepatitis virus-A59/RSA59, recombinant strain MHV-2/RSMHV2, Spike protein, fusion peptide (FP), fusion-mutant strain RSA59(P), RSMHV2(PP), cell-to-cell fusion (fusogenicity), neuroglial-tropism, neuropathogenesis, gliopathy

## Abstract

The β-Coronavirus mouse hepatitis virus (MHV-A59)-RSA59 has a patent stretch of fusion peptide (FP) containing two consecutive central prolines (PP) in the S2 domain of the Spike protein. Our previous studies compared the PP-containing fusogenic-demyelinating strain RSA59(PP) to its one proline-deleted mutant strain RSA59(P) and one proline-containing non-fusogenic non-demyelinating parental strain RSMHV2(P) to its one proline inserted mutant strain RSMHV2(PP). These studies highlighted the crucial role of PP in fusogenicity, hepato-neuropathogenesis, and demyelination. Computational studies combined with biophysical data indicate that PP at the center of the FP provides local rigidity while imparting global fluctuation to the Spike protein that enhances the fusogenic properties of RSA59(PP) and RSMHV2(PP). To elaborate on the understanding of the role of PP in the FP of MHV, the differential neuroglial tropism of the PP and P mutant strains was investigated. Comparative studies demonstrated that PP significantly enhances the viral tropism for neurons, microglia, and oligodendrocytes. PP, however, is not essential for viral tropism for either astroglial or oligodendroglial precursors or the infection of meningeal fibroblasts in the blood–brain and blood–CSF barriers. PP in the fusion domain is critical for promoting gliopathy, making it a potential region for designing antivirals for neuro-COVID therapy.

## 1. Introduction

The coronaviridae family is the largest known group of enveloped, non-segmented, positive-sense polarity RNA viruses. Coronaviruses (CoVs) are ubiquitously present and can infect a broad range of hosts, including humans and other mammals. Disease outcomes are diverse and related to the respiratory, enteric, hepatic, and central nervous systems [1,2]. The β-coronavirus genus poses a particular threat to humanity, as indicated by recent outbreaks of human CoV, including SARS-CoV-2, which caused the COVID-19 pandemic, as well as SARS-CoV and MERS-CoV, which caused limited, but lethal, outbreaks. Although the recent emergence of SARS-CoV-2 has elicited increased research on β-coronaviridae members, a lot remains to be done to further understand their pathogenesis. The mandatory requirement of biosafety labs to study these highly infectious viruses and a scarcity of suitable animal models warrant the use of closely related members that can be experimentally studied in limited resource settings.

Mouse hepatitis virus (MHV) is a murine β-coronavirus that has a long-standing history of being used to understand the pathogenesis of coronaviruses and associated disease manifestations based on the differential organ tropism of MHV strains [3,4,5,6,7]. Five different wild-type strains of MHV have long been well studied and reported. MHV-2 and MHV-3 are hepatotropic and can both induce meningitis, whereas MHV-3 can additionally cause encephalitis. A closely related strain of MHV-2 is MHV-A59, which possesses the hepatotropic and neurotropic ability to induce meningo-encephalomyelitis and demyelination concurrent with axonal loss. MHV-JHM is highly neurotropic, while MHV-1 induces respiratory syndrome [8,9,10,11,12]. MHV-A59 and MHV-2 share 91% identity in the genome and 83% identity in the pairwise Spike sequence [13], but they differ in their ability to induce cell-to-cell fusion. MHV-A59 is fusogenic, and MHV-2 is non-fusogenic.

Extensive prior studies revealed that the virus–host attachment Spike protein plays a significant role in viral fusogenic properties; infectivity; spread; and consequent neuropathologic manifestations such as optic neuritis, demyelination, and axonal loss [4,14,15,16]. A reverse genetic system for the coronavirus was leveraged, wherein individual structural proteins of the virion are targeted separately to understand their functional properties in pathogenesis. Recombinant strains RSA59 and RSMHV2, which are two isogenic strains except for the Spike gene, were engineered and used to demonstrate the role of the Spike protein in MHV pathogenesis. The recombinant strain RSA59 has the Spike gene of parental neurotropic demyelinating strain MHV-A59, and the recombinant strain RSMHV2 has the Spike gene replaced with the Spike gene of parental hepatotropic non-demyelinating strain MHV-2. In short, both viruses share the same genetic background of MHV-A59, except for their isogenic Spike protein [15,17]. Both the isogenic recombinant strains (RSA59 and RSMHV2) infect most neuronal cells, whereas only RSA59 follows infection with efficient cell-to-cell spread. RSA59 may spread through neurons, in between gray matter and white matter, following an axonal route where it can subsequently infect oligodendrocytes in the white matter due to its fusogenic ability [9,18]. Chronic effects of oligodendrocyte loss in the white matter induce moderate-to-severe demyelination. In contrast, RSMHV2, due to its inability to spread through neurons and its impaired fusogenic ability, is unable to reach and infect oligodendrocytes in white matter and thus is unable to induce demyelination. RSA59 can cause optic neuritis with the associated retinal ganglion cell loss due to its ability to follow retrograde transport along the optic nerve, whereas RSMHV2 is impaired in retrograde axonal transport through the optic nerve and thus is unable to cause optic neuritis or ganglion cell loss [16,19]. Previous studies have shown that the MHV-A59 Spike protein itself is capable of initiating the fusion process regardless of the presence of conventional MHV receptor CEACAM1 [20,21]. These studies reveal that the Spike protein is a central player in mediating cell-to-cell fusion and its consequent neuropathology.

In our attempt to delineate the minimum essential motif on the Spike gene required for fusogenicity and differential neuropathology, we targeted the fusion peptide (FP) of the Spike protein S2 domain due to its essential role in the fusion mechanism [22,23,24]. FP is one of the key functional regions on the Spike protein S2 subunit, which is essential for viral fusogenic properties both in the endosomal membrane and plasma membrane of the host cell [25,26]. At the early phase of the virus–host fusion process, the Spike protein remains unfolded and extended, which is followed by conformational changes where FP becomes anchored through the fusion core domain to the host plasma membrane [27,28]. FP is thus one of the targets for the development of antiviral molecules specifically as a fusion inhibitor for pan-coronavirus therapies, as it is conserved across coronaviruses and plays a significant role in the fusion mechanism [29,30,31,32,33,34,35,36]. We identified a central proline dyad (PP) in the FP domain that plays a critical function in the fusogenicity and hepato-neuropathology of MHV infection. Previously, we generated a proline deletion mutant RSA59(P), where one proline when deleted from the PP in the FP of RSA59(PP) significantly altered the viral fusogenicity, infectivity, antigen spread, and consequent neuropathology, including decreased demyelination and axonal loss. The alteration is due to the loss of structural rigidity provided by the PP dyad in the FP to the entire FP domain [21]. In another study, we found that optic nerve inflammation (optic neuritis) and consecutive myelin loss concurrent with retinal ganglion cell loss induced by RSA59(PP) infection is reduced in RSA59(P)-infected mice due to its inability to follow retrograde axonal transport to reach the retina through the optic nerve [37]. Recently, we generated two proline-containing mutant strain RSMHV2(PP), where we added one proline to the recombinant, non-fusogenic strain RSMHV2(P), which has one proline in the FP. Further, we investigated whether the introduction of PP in the FP alone can provide the adequate structural rigidity required for the efficient cell-to-cell fusion and hepato-neuropathology characteristic of RSA59(PP). The comparison between RSMHV2(PP) and RSMHV2(P) revealed that the addition of one proline significantly induces a fusogenic ability, viral antigen spread through neurons, and axonal transport to white matter that, in turn, causes discrete myelin loss. Necrotizing hepatitis caused by a RSMHV2(PP)-infected liver was significantly more than the RSMHV2(P)-infected liver at day 6 post-infection (p.i.). Distribution of the viral antigen in different neuro-anatomic regions in the infected brain of RSMHV2(PP) and RSMHV2(P) studied on days 3 and 6 p.i. clearly shows an increased ability of RSMHV2(PP) to penetrate brain parenchyma and cause enhanced encephalitis compared to RSMHV2(P) [38]. However, the degree of fusogenicity, neuropathology, and demyelination induced by RSMHV2(PP) does not match the extent of the pathology observed in RSA59(PP), as revealed by both in vitro and in vivo studies [38]. In silico studies suggested that structural rigidity posed by PP present in the FP in the fusion domain may be due to the transformation of the FP structure from helix-turn-helix-turn-helix in double proline-containing strains to a helix-loop-helix in single proline-containing strains. The local torsional flexibility generated by FP of RSA59(PP) and RSMHV2(PP) in simulations are different from RSA59(P) and RSMHV2(P), which show high dihedral fluctuations [21,38].

In the current study, we investigate the differential neuron and glial cell tropism by the two-proline and single-proline-containing strains to decipher the role of PP in the FP of the Spike protein of MHV on neurogliopathy. A comparison of RSA59(PP) with RSA59(P) and RSMHV2(P) with RSMHV2(PP) in in vitro and in vivo tropism studies of neurons, microglia, astrocytes, and oligodendrocytes reveals that the presence of double proline in the FP significantly enhances the tropism of neurons, as well as glial cells, microglia, and oligodendrocytes. However, the PP dyad in the FP is not crucial for astrocyte tropism and is also not essential for the infection of stem cells, precursors, or immature cells of oligodendrocyte lineage and meninges in primary cultures. These studies highlight the potential of the PP together with their neighboring residues in the FP for designing antivirals for pan-coronavirus-induced neuropathology, specifically gliopathy.

## 2. Materials and Methods

### 2.1. Viruses

The enhanced-Green Fluorescent Protein (EGFP) expressing isogenic recombinant fusogenic-demyelinating strain, RSA59(PP), and, fusion impaired-non demyelinating strain, RSMHV2(P) were used for deciphering the neuropathology mediated by Spike protein [5,9,18]. RSA59(P) and RSMHV2(PP) were engineered by deleting one proline from RSA59(PP) and inserting one additional proline to RSMHV2(P), respectively, through quick-change site-directed mutagenesis together with targeted RNA recombination in the center of the Spike protein’s FP region [15,21,23,38]. Briefly, the plasmids containing the desired mutation were isolated and linearized. An in vitro transcription kit was used to obtain synthetic mRNA. Targeted RNA recombination methods where synthetic mRNA acted as a donor and chimeric virus fMHV served as the recipient was used [17]. The candidate recombinant strains were selected and sequenced for confirmation of mutations performed as discussed in a recent publication [38]. The four variants of viruses used in the current study were parental and mutant recombinant strains of neurotropic MHV-A59 expressing EGFP, which only differ in the Spike gene. RSA59(PP) has the parental strain MHV-A59′s Spike gene, whereas RSMHV2(P) has the parental strain MHV2′s Spike gene, and mutants RSA59(P) and RSMHV2(PP) are the same as RSA59(PP) and RSMHV2(P), respectively, except for proline mutation in the FP.

### 2.2. Preparation of Mixed Glial Cultures from Day 0 Mouse Brain

Mixed glial cultures from day 0–1 pups of C57BL/6 mice were prepared following an established protocol from previous studies [39,40,41]. Briefly, whole brains were removed from pups, and meninges were carefully separated from brain tissues. Minced brain tissues in Hank’s balanced salt solution (HBSS) (Gibco, Waltham, MA, USA) containing trypsin (Gibco, Waltham, MA, USA) and DNase I (Thermo Scientific, Waltham, MA, USA) were incubated in a shaker water bath (at 100 rpm) at 37 °C for 30 min. Fetal Bovine Serum (FBS) (Gibco, Waltham, MA, USA) was added and samples were triturated, then washed and centrifuged at 2000 rpm for 10 min, after passing through a 70 µm strainer (Corning, Glendale, AZ, USA). The pellet was washed again with HBSS and spun down at 2000 rpm for 10 min before being diluted in astrocyte-specific Dulbecco’s Modified Eagle Medium (DMEM) (Gibco, Waltham, MA, USA) with 1% penicillin, 1% streptomycin (Gibco, Waltham, MA, USA), 1% non-essential amino acid (Gibco, Waltham, MA, USA), 1% L-glutamine (Gibco, Waltham, MA, USA), and 10% FBS. Cells were plated in a T25 flask and left for 24 h in a 5% CO_2_ humidified incubator at 37 °C. Non-adherent cells were washed off with HBSS and attached cells were maintained with in astrocyte-specific medium and grown till confluency with every 2–3 days media change.

### 2.3. Enrichment of Primary Astrocyte Cultures from Mixed Glia

Astrocytes were enriched from a confluent monolayer of mixed glia. Mixed glia were subjected to 10 days of starvation (stoppage of medium change) to allow differential adhesion of microglia and astrocytes. The cultured flask containing starved mixed glia was agitated in an incubator shaker at 37 °C, 200 rpm for 45 min, to enable weakly adhered cells to detach. The supernatant containing dislodged cells (mostly microglia) was collected for microglia enrichment. The adherent monolayer of cells after shaking were maintained in an astrocyte-specific medium and used as the astrocyte-enriched culture for experiments, with allowance of 2–3 passages with intact astrocyte morphology.

### 2.4. Enrichment of Primary Microglia from Mixed Glia

Primary microglia cultures were isolated following a protocol previously described [42]. Briefly, the supernatant obtained from agitated mixed glia post-starvation was immediately centrifuged at 200 rpm for 5 min, and the pellet (microglia) was resuspended in a fresh astrocyte-specific medium. Cells were counted using an automated cell counter (Countess II FL, Invitrogen, Thermo Scientific, Waltham, MA, USA) and plated either on a two-well CC2-treated chamber slide or Permanox (Nunc Lab-Tek chamber slide, Thermo Fisher Scientific, Waltham, MA, USA) coated four-well chamber slide. Microglia cells were allowed to attach and then media change was given to discard non-attached cells. Twenty-four-hour microglia cultures were then used for experimentation.

### 2.5. Enrichment of Oligodendrocyte Precursor Cells (OPCs) from Mixed Glia

Oligodendrocyte precursor cells (OPCs) were isolated from mixed glial cells prepared from neonatal day 0–1 mouse brains, as described previously [18]. Briefly, after removing non-adherent cells 24 h post mixed glial plating, adherent cells were washed with HBSS without Ca^2+^ and Mg^2+^. Fresh serum-free chemically defined medium containing growth factors (Neurobasal medium (Gibco, Waltham, MA, USA) containing B27 supplement (Gibco, Waltham, MA, USA) and 10 ng/mL of bFGF (Invitrogen, Waltham, MA, USA), 2 ng/mL PDGF (R&D Systems, Minneapolis, MN, USA), and 1 ng/mL NT-3 (Peprotech, Waltham, MA, USA)) was added. Cultures were maintained in this medium until confluency. Cells present in the mixed glial culture at that stage were predominantly astrocytes and OPCs. These cells were dislodged by a wash-down method to obtain OPCs that grow on top of astrocytes and were weakly adhered as opposed to the strongly attached astrocytes. Suspended OPCs were plated onto Poly-D-Lysine (Sigma, Saint Louis, MO, USA) coated coverslips and maintained in a serum-free chemically defined growth medium (OPCs growth medium) until the culture reached 80% confluency. The cells were then immunofluorescently characterized. The purity of the culture was determined using double staining with anti-A2B5 antibodies (markers for OPCs), anti-GFAP (a marker for astrocytes), anti-Olig2, anti-GalC (H8H9) (markers for mature oligodendrocytes), and anti-AP14 (a marker for mature neurons) together with DAPI counterstaining. OPC cultures of high purity were used for further experiments. 

### 2.6. Preparation of Primary Meningeal Fibroblast Cultures

Primary meningeal fibroblast cultures from day 0–1 old C57BL/6 pups’ brains were prepared as described in a previous study [40]. Briefly, the cranium was removed and intact brain tissue was exposed. The meninges were then carefully obtained and homogenized in HBSS. The homogenate was passed through a 70 μm cell strainer. The filtrate was collected and centrifuged at 1000 rpm for 10 min to obtain the meningeal cells as a pellet. The pellet was washed with HBSS two times and lastly resuspended in an astrocyte-specific medium as mentioned previously. Cells were plated and kept in a humidified 5% CO_2_ incubator at 37 °C. After 72 h, non-adherent cells were removed by washing with HBSS and fresh medium was added to adherent cells. The cultures were maintained as meningeal fibroblast cells in an astrocyte-specific medium until confluency with a media change every 2–3 days. The cultures were characterized using double immunofluorescence labeling with anti-vimentin (a pan-fibroblast marker), anti-aquaporin4 (a marker of differentiated fibroblasts), and anti-GFAP as well as DAPI counterstaining. Cultures were maintained in an astrocyte-specific medium and used for the experiment.

### 2.7. Propagation of Neuron-Enriched Primary Cultures Prepared from Day 0 Mouse Brains

Primary neuron-enrich cultures were prepared as previously described [38]. Briefly, brains were collected from neonatal pups (Day 0), and meninges were removed aseptically and entirely from whole brains. Brains were homogenized and incubated in a rocker water bath set at 37 °C for 30 min in HBSS, containing 300 µg/mL DNase I and 0.25% trypsin. Enzymatically dissociated cells were triturated in the presence of 0.25% FBS and centrifuged at 1500 rpm for 5 min. The pellet was resuspended in HBSS and passed through a 70 µm cell strainer. A second wash and centrifugation at 1500 rpm for 5 min was performed. Lastly, the resuspended cells were counted and diluted to 10^6^ cells/mL with DMEM containing 1% HEPES (Gibco, Waltham MA, USA), 1% penicillin, 1% streptomycin, 1% non-essential amino acid, 1% L-glutamine, and 10% FBS. Suspended cells were plated onto Poly-D-Lysin and Laminin (PDL/Lam) (Sigma, Saint Louis, MO, USA) treated culture plates and allowed to adhere for 24 h in a humidified chamber with 5% CO_2_ at 37 °C. After 24 h, non-adherent cells were removed and a fresh serum-free growth medium (Neurobasal medium containing B27 supplement, 1% penicillin, 1% streptomycin, and, 1% L-glutamine) was added. The cultures were then used for experimentation.

### 2.8. Infection of Primary Cultures with RSA59(PP), RSA59(P), RSMHV2(P), and RSMHV2(PP)

A monolayer of 80–90% confluent cells of primary cultures of neurons, microglia, astrocytes, oligodendrocyte precursor cells (OPCs), and meningeal fibroblast cells were infected with RSA59(PP), RSA59(P), RSMHV2(P), and RSMHV2(PP) at MOI 2 of inoculum prepared in the neurobasal medium for neurons and OPCs and DMEM containing 2% FBS for microglia, astrocytes, and meningeal fibroblast cells. Infected cells were incubated with the virus in a humidified incubator with 5% CO_2_ at 37 °C for 1 h 15 min with intermittent shaking at 15 min intervals for thorough viral adsorption. Inoculums were removed and the infected cells were given respective fresh maintenance media and kept in a humidified incubator with 5% CO_2_ at 37 °C. EGFP was monitored and images were taken at different time points post-infection for studying infectivity, syncytia formation, and viral antigen spread or immunofluorescent label for observing colocalization of EGFP with respective cell-specific markers.

### 2.9. Immunofluorescence on Primary Cultured Cells and Microscopy

Primary cultured cells for characterization, or infected primary cultures, were washed with 1X PBS containing Ca^2+^ and Mg^2+^ and fixed in 4% paraformaldehyde (PFA) and then washed with 1X PBS. For characterization and immunofluorescent labeling of infected OPCs and meningeal fibroblast cultures, cells were incubated with a blocking solution of PBS containing 0.5% of Triton X-100 and 2.5% goat serum (GS). Cells were then incubated with primary antibodies prepared (at 1:200 dilution) in the blocking solution for 1 h. The antibodies used with their source and dilutions are tabulated in Table 1. The primary antibody labeled cells were then washed thrice with 1X PBS containing Ca^2+^ and Mg^2+^ for 5 min and then incubated with the respective secondary antibodies conjugated with fluorophores prepared in blocking solution for 1 h, as indicated in Table 1. The cells were washed thoroughly with 1X PBS three times and carefully handled with minimal exposure to light and then mounted using Mowiol 4–88 (Sigma, Saint Louis, MO, USA) mounting media containing DAPI on glass slides. The slides were kept in the dark and allowed to dry before imaging. Images were acquired with a Nikon eclipse Ti2 epifluorescence microscope with a Nikon DS-Qi2 coupled camera (TYO, Japan) and confocal microscope (LSM710, Carl Zeiss, Jena, Germany), and processed with Zen black software (Carl Zeiss, Jena, Germany). Images were analyzed using Image J (Fiji, WI, USA) software v. 1.53.

### 2.10. Inoculation of Mice

Four-week-old MHV-free C57BL/6 mice were inoculated intracranially with half of the LD50 dose of 20,000 PFU for RSA59(PP), RSA59(P), and RSMHV2(PP), and 200 PFU for RSMHV2(P), as described in previous studies [5,17,18]. Mice were monitored daily for possible mortality and the signs and symptoms of the disease were recorded. Mock-infected groups received PBS with 0.75% BSA and were housed in the same conditions as the infected groups. All animals were euthanized by transcardial perfusion with PBS followed by 4% PFA at day 5 post-infection (p.i.) and brain and spinal cord tissues were harvested. For viral titer plaque assay, mice were perfused with 1X PBS, brain tissues collected in gel saline and stored in −80 °C until the assay was performed using routine viral plaque assay [43]. Three mice (N = 3) were used for each infection group alongside mock-infected mice for immunofluorescence analysis. The animal study protocol was approved by the Institutional Animal Ethical Committee of Indian Institute of Science Education and Research, Kolkata (protocol No. IISERK/IAEC/AP/2015/02.03, date of approval 13 July 2015).

### 2.11. Immunofluorescence in Brain and Spinal Cord Cryosections

Brain and spinal cord cryosections prepared from tissues harvested from day 5 p.i. mice were immunofluorescently labeled, as previously explained [5,18,38]. Briefly, sections were post-fixed with ice-cold 95% ethanol for 20 min and washed with PBS. Sections were then incubated in a humidified chamber, with 1M glycine (Sigma, MO, USA) prepared in PBS for 1 h followed by 1 mg/mL NaBH4 (SRL, Mumbai, India) for 10 min. Sections were blocked in blocking solution (PBS/2.5% GS/0.5% Triton-X100) for 1 h following a thorough PBS wash. Primary antibodies of different neuroglia markers with respective dilutions as indicated in Table 1 were prepared in a blocking solution and incubated overnight at 4 °C with the sections. The sections were thoroughly washed with PBS before incubating with secondary antibodies (as stated in Table 1) prepared in blocking solution for 1 h at room temperature in a humidified chamber. Sections were again washed with PBS and then mounted with DAPI containing Mowiol 4-88 mounting medium. Visualization of EGFP from infected cells by direct fluorescence microscopy and immunolabeled cells via the red channel was done using epifluorescent and confocal microscopes as mentioned in the previous section. 

### 2.12. Quantification of Colocalization In Vitro and In Vivo

Quantification of two or single proline-containing strains’ infectivity in vitro was evaluated by examining merged images of EGFP with MAP2, A2B5, and vimentin channels, respectively, for primary neurons, OPCs, and meningeal fibroblasts. About 20 frames in total of 20× magnification images were obtained from three independent experiments for each strain of the virus for analysis (Appendix A). For neurons, OPCs, and fibroblasts the amount of colocalization was determined as the Pearson’s coefficient using the JACoP plug-in feature of image J software where the pixel that contains the green and red fluorescence overlapping was automatically calculated. The DAPI staining colocalized with EGFP was used to calculate the percentage of infected primary microglia and astrocytes. Syncytia size in primary microglia was determined as previously discussed [38]. In vivo quantification of colocalization of specific stains with different neuroglia cells was determined by examination of 4–5 different regions in sagittal brain sections. Magnified images (40×) of EGFP and neuroglia-specific marker merged channel that shows relatively similar viral spread as well as a sufficient number of specific neural cells stained based on a random selection were quantified. The amount of colocalization in pixels was determined as stated for in vitro analyses. Percentages of infected microglia and astrocytes in culture were calculated. The amount of colocalization in microglia, astrocytes, and mature differentiated oligodendrocytes were estimated by ImageJ JACoP plug-in by Iba-1, GFAP, and ASPA staining, respectively, and EGFP-positive cells. To quantify oligodendrocyte infection in the spinal cord, about 10 random spinal cord sections were selected per mouse. White matter (dorsal, lateral, and ventral) of the spinal cord was examined for Olig2-positive staining that colocalized with EGFP. Two masked investigators analyzed these images following the same method as described above. 

### 2.13. Statistical Analysis 

Data were analyzed and plotted using GraphPad Prism 6.01 software. One-way ANOVA followed by Tukey’s multiple comparison test was used to compare RSA59(PP) versus RSA59(P), RSMHV2(P), RSMHV2(PP) and RSMHV2(P) versus RSMHV2(PP). The mean was presented in a violin plot and the *p*-value at or below 0.05 was considered statistically significant.

## 3. Results

### 3.1. Differences in Neuronal Tropism between RSA59(PP)-, RSA59(P)-, RSMHV2(P)-, and RSMHV2(PP)-Infected Primary Cultures and Mouse Brains

To understand the differential neural cell tropism of two- and single-proline-containing MHV strains, the primary neuronal cultures [38] were infected with two proline-containing MHV (RSA59(PP) and RSMHV2(PP)) and single proline-containing MHV (RSA59(P) and RSMHV2(P)). After 24 h of infection at a MOI of 2, cultured cells were immunostained with anti-MAP2 antibody and counter-stained with DAPI. The colocalization of neurons and viral antigen was compared between RSA59(PP) versus RSA59(P) and RSMHV2(P) versus RSMHV2(PP)-infected cells. The colocalization of EGFP, the intrinsic viral antigen reporter, in MAP2-positive neurons showed that the four variants of MHV infected neurons with differential efficiency (Figure 1). RSA59(PP) showed the highest efficiency of neuronal tropism in the primary neuron culture (Figure 1A–D). There was a significant decrease in the number of neurons infected by RSA59(P) when compared to RSA59 (PP) (Figure 1E–H). Similarly, RSMHV2(PP) efficiently infected neurons, and the infection was significantly reduced in RSMHV2(P)-infected cultures (Figure 1I–P). As previously reported [38], fusogenicity and neuronal spread were significantly observed in RSA59(PP) and RSMHV2(PP)-infected cultures. In contrast, RSA59(P) induced much less neuronal spread of the viral antigen. RSMHV2(P) showed the single-cell infection of majorly non-neuronal cells in the culture, as shown by a large number of EGFP-positive cells, but reduced the colocalization with MAP2-positive cells (Figure 1I–L). The quantification data of differential neuronal colocalization with the viral antigen was plotted in a violin diagram (Figure 1Q). 

Next, the differential ability of the four variants to infect neurons in vivo was evaluated. Prior studies showed that MHV replication following intracranial inoculation reaches its peak at day 5 p.i., and the peak of inflammation occurs at day 7 p.i. Between days 5 and 7 p.i., viral spreading with limited cellular damage has been reported [9,42]. Therefore, we harvested brain tissues from mice at day 5 p.i. and processed them for cryosections. Direct fluorescence microscopy was used to visualize EGFP in the infected tissue sections. Sagittal brain cryosections were immunofluorescent-labeled with anti-MAP2 antibody using a red fluorescent AlexaFluor 568-conjugated secondary antibody. Visual observation of the microscopic images of merged channels revealed that RSA59(PP) infected neurons most efficiently. There is a decrease in the number of neurons infected by RSA59(P) when compared to RSA59(PP). Similarly, RSMHV2(PP) showed an increase in neuronal tropism compared to RSMHV2(P) (Figure 2). Though RSMHV2(P) showed many EGFP-positive cells, very few were positive for neuronal markers (Figure 2). The amount of colocalization assessed as a Pearson coefficient was plotted in a violin diagram (Figure 2). In summary, the results demonstrated that the presence of PP in the FP of the Spike protein of MHV significantly enhanced the ability of MHV to infect neurons. This study demonstrated the crucial role of PP in the Spike protein FP of murine β-coronavirus MHV in neuronal tropism in vivo. The viral titration of brain tissue following the intracranial inoculation of RSA59(PP), RSA59(P), RSMHV2(P), and RSMHV2(PP) at day 5 and day 7 p.i. revealed significant differences in the replicating efficiency of the four virus strains. The two proline-containing strains RSA59(PP) and RSMHV2(PP) were capable of robust viral replication as compared to the single proline-containing strains RSA59(P) and RSMHV2(P) (Figure 2V). 

### 3.2. RSA59(PP), RSA59(P), RSMHV2(P), and RSMHV2(PP) Differ in Microglial Tropism Both In Vitro and In Vivo

The microglial tropism of RSA59(PP), RSA59(P), RSMHV2(P), and RSMHV2(PP) was examined in 95–98% CD11b-positive microglia cultures (Appendix A) [42]. EGFP observed from direct fluorescence microscopy of cultures was monitored and imaged from 8 h p.i. until 24 h p.i. when the cells were counterstained with DAPI. Timed kinetic studies revealed that the four variants could infect microglia as early as 8 h p.i. RSA59(PP) started to form syncytia at 8 h p.i. with a profuse infection that increased with time, whereas RSA59(P) showed delayed syncytia formation only observed at 12 h p.i. that was comparably fewer in number compared to RSA59(PP)-infected cultures. However, RSMHV2(P) and RSMHV2(PP) did not exhibit syncytia formation at 8 h p.i., but efficient single-cell infections were observed. Clusters of infected cells were observed in RSMHV2(PP) infected cultures at 12 h p.i. (Appendix A). At 24 h p.i., syncytia in RSA59(PP) infected cultures began to dissolve, whereas the few syncytia in RSA59(P) infected cultures did not dissolve and remained fewer in number compared to RSA59(PP) (Figure 3A–H). We quantified the size of the syncytia by measuring the number of nuclei per syncytium. There was no significant difference in the size of syncytia formed by RSA59(PP) and RSA59(P) at 24 h p.i. However, the number of infected microglia in RSA59(PP) infected cultures was more than that of RSA59(P) infected cultures at any time point p.i. RSMHV2(P) and RSMHV2(PP) infected cultures even at 24 h p.i. did not form syncytia (Figure 3I–P). Interestingly, the percentage of EGFP-positive cells indicating infected microglia in the culture demonstrated a significant increase in response to RSA59(PP) compared to RSA59(P), and in RSMHV2(PP) compared to RSMHV2(P) (Figure 3Q). Increased fusogenic ability was observed in RSMHV2(PP) infected cultures as clusters of infected cells were observed in contrast to their absence in RSMHV2(P) infected cultures (Figure 3P). 

In vivo study of microglial tropism revealed that all four variants can infect microglia in the brain as colocalization of Iba-1-positive microglia cells (Red) and viral antigen (Green) in infected cells was seen in all groups of infected mice as compared to mock-infected mice (Figure 4). There was a pathomorphological difference in stained microglia observed among the four variants in infected brain sections. RSA59(PP) and RSA59(P) showed predominantly phagocytic microglia, whereas RSMHV2(P) and RSMHV2(PP) showed ramified microglia in the brain parenchyma, and mock-infected brain sections showed resting microglia (Figure 4A–T). Detailed colocalization studies of Iba-1-positive cells with viral antigen and quantification of the amount of colocalization showed an increase of microglial tropism in RSA59(PP) infected mice compared to RSA59(P). Similarly, RSMHV2(PP) caused a significant increase in infected microglia compared to RSMHV2(P). The results show a highly preferential microglial tropism by PP-containing MHV strains as opposed to single proline-containing MHV strains that exhibit significantly less preference for microglial infection both in vitro and in vivo (Figure 4U).

### 3.3. RSA59(PP), RSA59(P), RSMHV2(P), and RSMHV2(PP) Differ in Their Infectivity, and Syncytia Formation in Primary Astrocyte Cultures, and Ability to Induce Astrogliosis In Vivo

Previously characterized astrocyte enriched cultures were infected to examine differential astrocyte tropism by PP-containing MHV strains and single proline containing strains, at an MOI of 2 [39,44]. Direct fluorescent imaging of EGFP from infected cells was performed on the cultures at a regular interval of 6 h p.i. and monitored for 48 h p.i. The four strains infected astrocytes at 12 h p.i. However, the infection started earlier in the RSMHV2(P) infected cultures compared to other strains and other neuroglial cells. RSA59(PP) started forming syncytia at 24 h p.i., which increased in size with time, whereas RSA59(P) infected cultures did not show syncytia formation at all (Figure 5B,E). At 36 h and 48 h p.i., giant syncytia were observed in RSA59(PP) infected cultures (Appendix A). RSMHV2(P) and RSMHV2(PP) showed robust single-cell infection at 24 h p.i. (Figure 5H,K) where a progressive increase in infected cells by RSMHV2(P) was more than that observed in RSMHV2(PP) infected cultures even at 36 h and 48 h p.i. (Appendix A). Quantification of EGFP-positive cells in DAPI counterstained images at 24 h p.i. revealed a significant increase in RSA59(PP) compared to RSA59(P). In contrast, RSMHV2(P) showed the same percentage of infectivity as observed with RSA59(PP) and was significantly increased as compared to RSMHV2(PP). Data were plotted and presented in a violin diagram (Figure 5M). 

To further confirm the observations of differential infectivity of RSMHV2(P) with PP-containing strains of RSMHV2(PP), we examined the astrocyte tropism in mouse brain cryosections at day 5 p.i. Morphological observation of the red channel (GFAP-positive cells) showed reactive astrocytes in the infected groups, whereas resting/non-reactive astrocytes were seen in mock-infected mice (Figure 6). RSA59(PP) and RSA59(P) infected mice showed fewer EGFP-positive cells in regions with numerous GFAP-positive cells (Figure 6E,I). In contrast, RSMHV2(P) and RSMHV2(PP) infected mice showed relatively high numbers of EGFP-positive cells in regions with a large number of GFAP-positive cells (Figure 6M,Q). Astrogliosis, a relatively high number of reactive astrocytes, was observed in RSMHV2(P) infected mice (Figure 6N). Colocalization of anti-GFAP staining with EGFP revealed that RSMHV2(P) induced the highest amount of colocalization of the virus within astrocytes (Figure 6O). Non-neuronal EGFP-positive cells observed in Figure 2R were likely astrocytes. RSA59(PP) showed an increase in the number of astrocytes infected compared with RSA59(P), but the degree of colocalization of infected astrocytes by RSA59(PP) was less when compared to RSMHV2(P). The number of RSMHV2(PP) infected astrocytes was significantly less compared with RSMHV2(P) (Figure 6U). Taken together, these observations suggest that the PP in the FP of the Spike protein of MHV was not critical for efficient astrocyte tropism in astrocyte cultures or in brain tissues at the acute stage of infection. The current study, to the best of our knowledge, is the first to reveal that presence of a single proline, and not necessarily PP in the center of the FP of the Spike protein of β-coronavirus MHV, is capable of inducing efficient astrocyte tropism in comprehensive in vitro and in vivo studies.

### 3.4. Differential Ability of RSA59(PP), RSA59(P), RSMHV2(P), and RSMHV2(PP) to Infect Oligodendrocyte Precursors (OPCs) In vitro and Myelinating Oligodendrocytes In Vivo

Oligodendrocytes are important glial cells for myelination, their viral infection is detrimental to brain function. To determine the role of PP in the FP of the Spike protein of MHV on oligodendrocyte tropism, oligodendrocyte precursor cells were enriched and cultures were prepared. Visual manual quantification of the immunolabeled cultures by counting revealed that 98% of the cells in the cultures were A2B5-positive, indicating that the culture was enriched with oligodendrocyte precursors (Figure 7A). Very few cells matured and stained positive for Olig2, accounting for 5% of the cell population, whereas less than 2% of the cells in the cultures were neurons or astrocytes, as shown by the violin plot (Figure 7D). 

OPCs cultures were infected with RSA59(PP), RSA59(P), RSMHV2(P), and RSMHV2(PP) at MOI 2. After 24 h p.i., OPCs got infected with the four variants of the virus. RSA59(P) showed reduced infection compared to RSA59(PP) (Figure 7E,I). Surprisingly, RSMHV2(P) robustly infected OPCs significantly more than RSA59(PP) and RSMHV2(PP) (Figure 7M). We previously reported similar OPCs infection by RSA59(PP) and RSMHV2(P) [18]. EGFP-positive cells that were double positive for A2B5 were manually counted and plotted (Figure 7U). 

Oligodendrocyte tropism was determined in mouse brain tissues following intracranial inoculation of the four variants into the mice, at day 5 p.i. Cryosections of mouse brains sectioned sagittally were immunofluorescently labeled with anti-ASPA, a marker of oligodendrocytes. EGFP from infected cells in the brain parenchyma and ASPA-positive cells was visualized via green and red channels, respectively. Colocalization of EGFP with ASPA staining indicated infected oligodendrocytes. Scanning of the brain sections showed a similar pattern of oligodendrocytes distributed in different neuroanatomic regions (Appendix A). RSA59(PP) infected mice showed an increase in the number of infected oligodendrocytes compared to RSA59(P) infected mice. RSMHV2(PP) infected mice induced a similar oligodendrocyte infection rates compared to RSMHV2(P), though not statistically significant (Figure 8U). 

To further validate the outcome observed in RSMHV2(PP) infected brain sections, oligodendrocyte tropism of the four variants of the viruses was examined in the spinal cord on day 5 p.i. Olig2 primary antibody that stains the nucleus of oligodendrocytes was used to label oligodendrocytes (Red). Fluorescence from EGFP denoting viral antigen (Green) was present in both gray and white matter of mice infected with RSA59(PP), RSA59(P), and RSMHV2(PP), whereas RSMHV2(P) showed EGFP in the gray matter only, the same as previously reported [38]. Olig2-positive cells were distributed in both gray and white matter with relatively high density at the dorsal horn gray matter in all groups of infected mice and mock-infected mice (Figure 9). Colocalization revealed that RSA59(PP) caused an increase in oligodendrocyte tropism when compared to RSA59(P). Similarly, RSMHV2(PP) showed robust infection of oligodendrocytes that was more than RSA59(PP) induced oligodendrocyte tropism. Since only white matter was considered, there were no infected oligodendrocytes observed in the white matter of RSMHV2(P) infected mice (Figure 9M–P) due to restricted infection of RSMHV2(P) to gray matter. In summary, PP-containing strains of MHV induced oligodendrocyte tropism more efficiently than single proline-containing strains in vivo (Figure 9U). Thus, MHV infection of myelinating oligodendrocytes, but not its precursors, is controlled by the presence of PP in the FP of the Spike protein.

### 3.5. Differential Ability of RSA59(PP), RSA59(P), RSMHV2(P), and RSMHV2(PP) to Infect Primary Meningeal Fibroblast Cells in Culture and Choroid Plexus Fibroblasts in Mouse Brain

The ability of double proline and single proline-containing strains of MHV to cause infection was investigated in previously established primary meningeal fibroblast cell cultures [40]. Immunofluorescence characterization of primary meningeal fibroblast cultures showed almost all cells positive for vimentin and a few cells double positive for vimentin and GFAP (Figure 10A). Aquaporin4, a marker for differentiated fibroblasts, stained many cells in the culture (Figure 10B). Visual counting revealed that 98% of cells in the culture were vimentin-positive with 16% GFAP-positive and about 80% Aquaporin4-positive cells as shown in the violin graph (Figure 10D). Heterogeneous populations of cells were observed in the cultures even though they all stained positive for vimentin. This suggests that stem/progenitor cells that were capable of becoming fibroblasts and or other cells such as astrocytes, ependymal cells, and pericytes were present in the culture. To determine the effect of PP on the MHV infection of meningeal fibroblasts, RSA59(PP), RSA59(P), RSMHV2(P), and RSMHV2(PP) were used to infect meningeal fibroblast cultures at MOI 2. After 24 h p.i., cells were immunofluorescently labeled with vimentin and visualized. RSA59(PP) and RSA59(P) showed similar infectivity in primary meningeal fibroblast cultures (Figure 10E–L). The infectivity was robust in cultures infected with RSMHV2(P), and the RSMHV2(PP) infected cultures showed more infection than the RSA59 groups. However, a detailed colocalization study revealed that stem or progenitor cells were predominantly infected by RSA59(PP) and RSA59(P), whereas RSMHV2(P) and RSMHV2(PP) infected both differentiated meningeal fibroblasts and stem cells in the culture (Figure 10U).

The heterogeneity of primary meningeal fibroblast cultures made it difficult to examine the tropism for specific fibroblast isotypes by these MHV strains. Thus, we determined fibroblast tropism by the four variants at day 5 p.i. in cryosections of mouse brains. Systematically scanned sagittal brain sections of the red channel showed that vimentin-positive cells, which were believed to be fibroblasts, were predominantly present in meninges, choroid plexus, and the subventricular zone, with a similar pattern in both infected and control mice. Detailed colocalization study of EGFP and vimentin stained cells demonstrated that RSA59(PP) and RSA59(P) both could efficiently infect fibroblast cells of choroid plexus (Figure 11C,G), whereas RSMHV2(P) showed much less ability to infect choroid plexus fibroblasts (Figure 11K). RSMHV2(PP) showed an increase in choroid plexus fibroblast tropism when compared to RSMHV2(P), but it was less than that observed in the case of RSA59(P) and RSA59(PP) infection. The results from primary meningeal fibroblast cultures and in vivo studies suggest that the efficiency of brain fibroblast tropism by MHV strains do not rely on the presence of PP in the FP of its Spike protein. The proline–proline dyad in the FP of the Spike protein was not important for MHV infection of meningeal/choroid plexus fibroblasts in regions of the blood-brain barrier and the blood-CSF barrier (Figure 11Q). Overall, the degree of neuroglial tropism in two proline containing strain, RSA59(PP), and RSMHV2(PP) and single proline containing strain, RSA59(P), and RSMHV2(P) as investigated in vitro and in vivo is graphically depicted in Figure 12. 

## 4. Discussion

Enveloped viruses such as the coronavirus possess a Spike fusion protein that mediates the virus–host attachment and the subsequent membrane fusion mechanism. The FP, which is characterized by a hydrophobic stretch with the propensity to interact with a biological membrane, is one of the conserved functional regions in the Spike protein of coronaviruses. Our previous studies identified PP to be a minimum essential motif on the FP of the Spike protein of MHV-A59/RSA59 and MHV-2/RSMHV2. The presence of PP in the FP controls the fusogenicity, viral antigen spread in vitro and in the CNS, and retrograde axonal transport of the viral antigen [21,37,38], but little is known about its neuroglial tropism. The current study highlights a crucial role of PP underlying neuroglial tropism. Studies have compared the PP-containing parental RSA59(PP) and RSMHV2(PP) mutant with single proline-containing strains RSA59(P) mutant and parental RSMHV2(P), respectively, in primary cultures enriched in neurons, microglia, astrocytes, oligodendrocyte precursors, and meningeal fibroblast cells. Studies were further extended to in vivo colocalization studies of infected cells with the individual markers of neurons and glial cells. Two proline MHV strains RSA59(PP) and RSMHV2(PP) show higher neuronal tropism with higher viral antigen spread in neuronal cells in mouse brains as compared to single proline strains RSA59(P) and RSMHV2(P). This result shows the importance of PP in FP in enhancing neuronal tropism and neurovirulence and is consistent with previous findings [5,45], where strain SJHM-RSA59 that has PP in FP of its Spike protein also induced higher neurovirulence.

Microglial tropism in the primary culture following infection demonstrates an increase in RSA59(PP) compared to RSA59(P) and RSMHV2(PP) compared to RSMHV2(P). A similar trend was observed in in vivo studies of infected brain sections stained with microglia markers, where PP-containing strains infected more microglia than their corresponding single P-containing strains. Previous studies have demonstrated microglia are one of the key players in neuroinflammation induced by different strains of MHV [9,42,46]. Optic nerve inflammation and RGC loss observed in RSA59(PP)-infected mice are more than those seen in RSMHV2(P) [19] and in single proline-deleted mutant RSA59(P) [37]. Our previous study that compared neuroinflammation in the brains and spinal cords of mice infected with RSA59(PP) and RSA59(P) at day 3 and 6 p.i. reported no significant difference in microglial activation [21]. However, the current study not only examined the microglial activation but also the microglial tropism by colocalizing with the viral antigen, which was not assessed in the previous study. 

Unlike neurons and microglia, the presence of PP in FP appears not to be essential for efficient astrocyte tropism. A single proline in FP was sufficient to induce astrogliosis with a concurrent tropism of astrocytes (Figure 6). Thus, the PP dyad has no significant role in astroglial tropism. That could be due to the differential mechanisms of viral entry and tropism in different neuroglial cells. In addition, the presence of PP in RSMHV2(PP) could neither increase the percentage of infected astrocytes to match RSMHV2(P) nor form the syncytia that are observed in RSA59(PP). Similar results were observed in a previous study where a higher astrocyte infection in RSMHV2(P)-infected brain sections was reported compared to RSA59(PP) and SJHM-RSA59-infected brain sections [5]. 

The current study also revealed a differential oligodendrocyte tropism by PP-containing strains compared to single proline-carrying strains both in vitro and in vivo, demonstrating that the presence of PP in FP of the Spike protein of MHV promotes the ability of the virus to infect oligodendrocytes. The myelinating function of oligodendrocytes is predominantly observed in white matter; thus, studying the colocalization of viral antigens with mature oligodendrocytes in white matter is warranted. A major challenge lies in demarcating gray and white matter in the mouse brain. Due to the clear demarcation of the gray and white matter of the spinal cord, spinal cord samples were chosen for colocalization studies of mature oligodendrocytes (Olig2) with viral antigens. The results indicated that oligodendrocyte tropism is increased by the presence of PP in the FP of the MHV Spike protein. In vitro experiments of oligodendrocyte tropism in our previous study [18] reported a similar differential ability of RSA59(PP) and RSMHV2(P) to infect oligodendrocytes. The present study also investigated the role of PP on OPC tropism using two additional Spike protein mutants: RSA59(P) and RSMHV2(PP). Surprisingly, RSMHV2(P)-infected OPC cultures showed a significant increase in OPC infection. The significant difference obtained in the current study might be due to a higher percentage of OPCs with fewer astrocytes in the culture, as well as lower MOI (one is used), which caused an increased infection of OPCs by RSMHV2(P), a trend similar to astrocyte culture infections by the four variants. Being a precursor cell, OPCs may have astrocytic properties at this stage of differentiation. The presence of PP enhanced the infection of mature oligodendrocytes, which may partly explain why RSA59(P) shows a reduced demyelinating ability in the spinal cord [21] and optic nerve [37]. Further, RSMHV2(PP) could induce myelin loss and mild demyelination [38], whereas SJHM-RSA59 and RSA59(PP) caused moderate-to-severe demyelination. 

Primary meningeal fibroblast cell cultures infected with the four virus strains revealed that the presence of PP in FP is not essential to effectively infect meningeal fibroblasts. RSA59(PP) infects a similar number of fibroblasts as compared to RSA59(P), while the infection rate is significantly higher with RSMHV2(P) as compared to RSMHV2(PP). This result is similar to the astrocyte tropism observed in vitro and in vivo and is consistent with observations that wild-type MHV-2 does cause meningitis [4,8], most likely because the central single proline in its FP is sufficient to mediate viral fusion with meningeal cells. However, in vivo studies of the fibroblast tropism in choroid plexus revealed a similarly efficient infection of fibroblast-like cells in the choroid plexus of RSA59(PP) and RSA59(P)-infected brains, whereas RSMHV2(P) caused a limited viral spread in the choroid plexus, and interestingly, RSMHV2(PP) showed a significantly increased fibroblast tropism compared to RSMHV2(P). Penetration of the viral antigen deep into the neuroglial cells of the brain parenchyma is essentially controlled by the presence of PP, as reported in our previous studies [3,21,38]. 

Overall, the strategic position of the FP together with its PP in the Spike protein, and the ability to control fusogenicity, hepato-neuropathogenesis, demyelination, and neurogliopathy, as revealed in the current study, makes FP a suitable target region in the coronavirus genome for the development of antivirals, possibly in the form of mimetic peptides. Amino acid residue mutagenesis studies on the FP region of the La crosse virus have shown attenuation in neuropathogenesis, specifically differential neuroinvasion and neurovirulence of the mutant virus [32,47,48]. In addition, mounting evidence from related strains of viruses such as HIV, reovirus, and TMEV that are capable of inducing neuroinflammation support the postulation that different CNS cell tropism and spread in the CNS play a key role in neurodegeneration [49,50]. Thus, understanding the genomic control of neuroglial tropism in MHV as investigated in the present study may help in tackling neurodegeneration induced by the coronaviruses. Combining in vitro and in vivo experiments that evaluate the differential tropism of PP and single proline-containing strains of MHV on the neurons and major glial cells provides insights into the preferential infection of different CNS cell types. The results suggest this differential tropism could be due to the presence of PP in the center of FP of the Spike protein of MHV, which provides the requisite rigidity. These insights are relevant in understanding the mechanism employed by the coronavirus–cell fusion process and may lead us towards developing therapeutics to prevent neuro-COVID. The Spike protein has a unique trimeric structure where the activation of a viral fusion requires an irreversible conformational transition of its S2 domain to a trimeric helical-bundle state. The FP is juxtaposed in the prefusion structure at a location that favors early contact initiation with the virus membrane [51], while a lower conformational transition barrier of the S2 domain facilitates the membrane fusion, and the absence of the dominant transition paths may lead to many aborted fusion events. PP in the Spike protein, by inducing local rigidity, makes the early contact points of the Spike protein more stable and likely enforces the conformational transition paths that are less abortive. The virus fusogenicity would thus be guided by the membrane composition of the host cell, and the fusion pathway utilized, i.e., direct entry or the endosomal pathway, thereby determining their cellular tropism.

Furthermore, the additional proline has a structural role in altering the local conformation by inducing helix-loop conformation in the polypeptide neighborhood. At the same time, it alters the global flexibility of the Spike S2 domain to allow an easier transition to the post-fusion conformation, contributing to the accelerated kinetics of the fusion process [21,38]. The ability of proline (an amino acid) to induce cis-trans-isomerization of the peptide bond, which was noted in another study [52], could additionally contribute to the enhanced fusogenicity of the Spike S2 domain by exposing the buried hydrophobic residues through a geometric reorganization that facilitates the rupture of the host membrane. The differential fusion activities of RSA59(PP) and RSMHV2(PP) compared to RSA59(P) and RSMHV2(P) thus drive an altered pathogenesis that can be discerned from the tropism studies, where cell-to-cell fusion plays a key role in viral spreading, ensuring an acute or chronic infection. The current study reiterates the importance of the proline dyad in enhancing the viral tropism for the neurons, microglia, and oligodendrocytes. Single proline-containing RSMHV2(P) cannot trigger demyelination despite infection [3,15], and its inter-neuronal spread is limited in vivo and in vitro due to a lack of translocation from gray matter to white matter in the spinal cord [18]. The axonal loss is restricted to early localized neurodegenerative changes only [9]. In contrast, RSA59(PP) aggressively spreads infection through the neuroglia in a retrograde and anterograde manner [37]. These suggest a definitive role of proline dyad in the fusion activities of RSA59(PP) and RSMHV2(PP) compared to RSA59(P) and RSMHV2(P) vis-à-vis steering the outcomes of pathogenesis.

## 5. Conclusions

The proline–proline dyad in the center of FP of the Spike protein of β-coronavirus plays an essential role in the neuropathogenesis and neurogliopathy induced by coronavirus infection. PP-containing strains RSA59(PP) and RSMHV2(PP) preferentially infect neurons, microglia, and oligodendrocytes efficiently, whereas single proline-containing strain RSMHV2(P) efficiently infects astrocytes, precursor cells of oligodendrocytes and meninges, indicating that the presence of double proline in FP modulates neuronal tropism and gliopathy. Overall, PP is responsible for differential neuroglial tropism and significantly contributes to properties such as viral antigen spread, cell-to-cell fusogenicity, viral entry, and its consecutive neuropathogenesis. Parental RSMHV2(P) has the limited ability to infect neurons, is unable to spread from gray to white matter, and is successively impaired in infecting white matter oligodendrocytes, and thus, it cannot cause demyelination. This entire mechanistic aspect can be clearly explained by the current neuroglial tropism studies of the four different mutants, answering the long-standing question of why RSMHV2(P), although very closely similar to RSA59(PP), differed in its ability to invade the white matter and cause demyelination. 

## Figures and Tables

**Figure 1 viruses-15-00215-f001:**
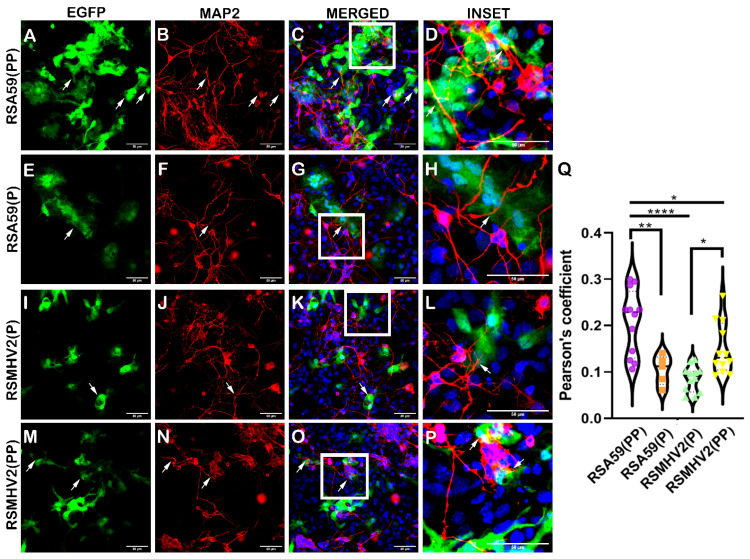
Infection of primary neuron enriched cultures with parental strains RSA59(PP), RSMHV2(P), one proline deleted RSA59(P), and one proline added RSMHV2(PP) mutants. Primary neuronal cultures isolated from day 0 C57BL/6 mice were infected with RSA59(PP) (**A**–**D**), RSA59(P) (**E**–**H**), RSMHV2(P) (**I**–**L**), and RSMHV2(PP) (**M**–**P**) at MOI 2. Cells were immunolabeled with MAP2 antibody at 24 h p.i. and counterstained with DAPI. Images presented show infected cells in the EGFP panels, and MAP2 panels indicate neurons in the culture. Merged panels show neurons with EGFP colocalization, with DAPI indicating the nucleus and arrowheads indicating infected cells. Panels (**D**,**H**,**L**,**P**) show colocalization in insets of higher magnification highlighted by a white rectangle area in the merged images. Quantified colocalization was presented in a violin diagram as the Pearson coefficient (**Q**). Experiments were performed three times in triplicate per strain (N = 9). The level of significance was taken at *p* < 0.05 following One-way ANOVA analysis. * *p* < 0.05, ** *p* < 0.001 and **** *p* < 0.00001.

**Figure 2 viruses-15-00215-f002:**
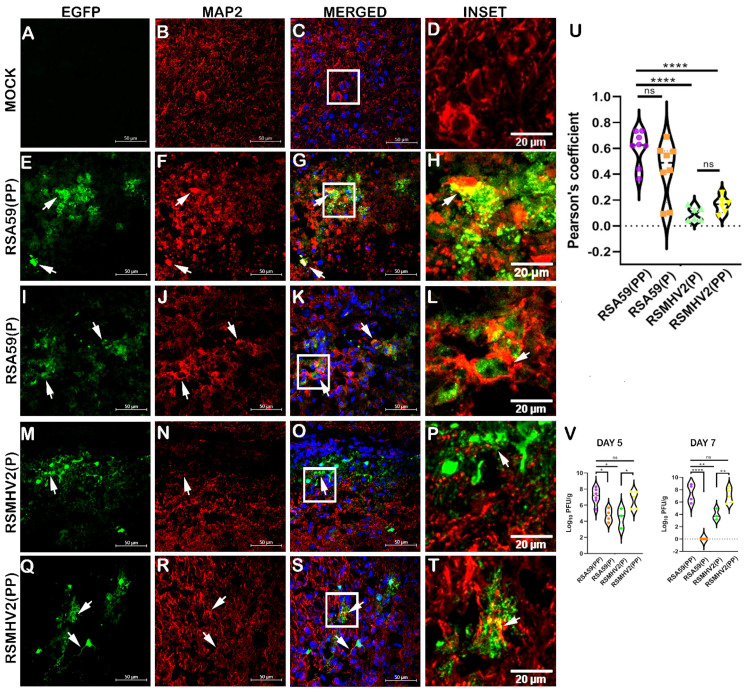
Colocalization study of two proline containing strains RSA59(PP), RSMHV2(PP) and single proline containing strains RSMHV2(P), RSA59(P) infected neurons in the brain of mice at day 5 p.i. Infected mice alongside control mock-infected mouse brain cryosections were immunofluorescently labeled with MAP2 antibody (**B**,**F**,**J**,**N**,**R**). EGFP panels indicate infected cells in the tissue (**A**,**E**,**I**,**M**,**Q**). Panels (**C**,**G**,**K**,**O**,**S**) show merged images of EGFP and MAP2 staining with arrowheads highlighting virus infected neurons. RSA59(PP) infected sections are in panel (**E**–**H**), RSA59(P) infected sections are in panel (**I**–**L**), RSMHV2(P) infected sections are in panel (**M**–**P**), RSMHV2(PP) infected sections are in panel (**Q**–**T**), and mock-infected sections are in panel (**A**–**D**). The insets marked by a white rectangle area in the merged panels show a magnified view of the colocalization of neurons with EGFP (seen as yellow) in panels (**D**,**H**,**L**,**P**,**T**). The amount of colocalization quantified as the Pearson coefficient was plotted in a violin diagram (**U**). Viral titer was performed at day 5 and day 7 p.i. to compare virus replication among RSA59(PP), RSA59(P), RSMHV2(P), and RSMHV2(PP). The results were plotted as a violin plot (**V**). The One-way ANOVA was used to compare RSA59(PP) with RSA59(P), RSMHV2(P), RSMHV2(PP), and RSMHV2(P) with RSMHV2(PP) infection. The level of significance was taken at *p* < 0.05, * *p* < 0.05, ** *p* < 0.001, **** *p* < 0.00001, and ns denotes non-significance.

**Figure 3 viruses-15-00215-f003:**
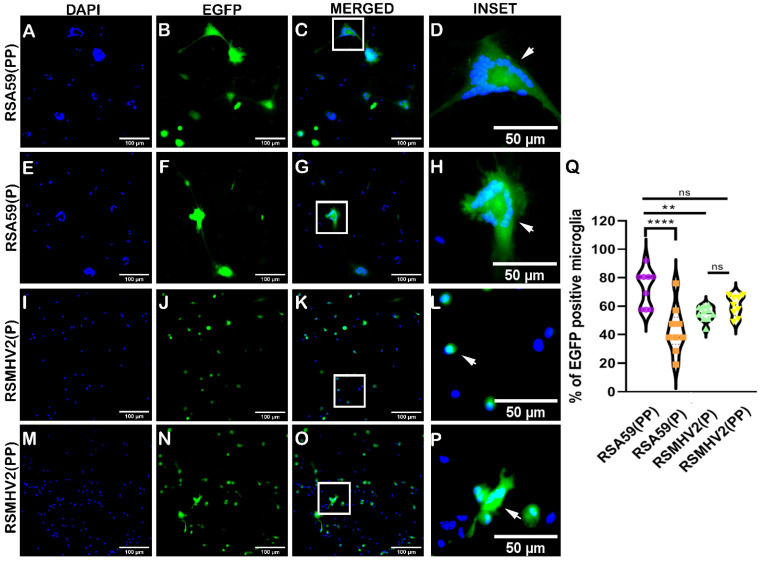
Differential infectivity and syncytia formation in primary microglia cultures infected with parental strains RSA59(PP), RSMHV2(P), one proline deleted RSA59(P), and one proline added RSMHV2(PP) mutants. Primary microglia cultures enriched from mixed glial cultures prepared from day 0–1 C57BL/6 mice pups were infected at MOI 2 with RSA59(PP) (**A**–**D**), RSA59(P) (**E**–**H**), RSMHV2(P) (**I**–**L**) and RSMHV2(PP) (**M**–**P**). Cells were counterstained with DAPI at 24 h p.i. Representative images show infected cells in the EGFP panels. Merged panels show the nuclei of infected cells and syncytia in insets marked by a white rectangle area in panels (**C**,**G**). A magnified view of insets marked by a white rectangle area is shown in panels (**D**,**H**,**L**,**P**). Percentages of infected cells in RSA59(PP), RSA59(P), RSMHV2(P) and RSMHV2(PP) infected cultures were quantified and plotted in a violin diagram (**Q**). The white-colored arrowheads in Insets panel indicates a cluster of infected cells. Experiments were performed three times in triplicate per strain (N = 9). The level of significance was taken at *p* < 0.05 following One-way ANOVA analysis. ** *p* < 0.001, **** *p* < 0.00001, and ns denotes non-significance.

**Figure 4 viruses-15-00215-f004:**
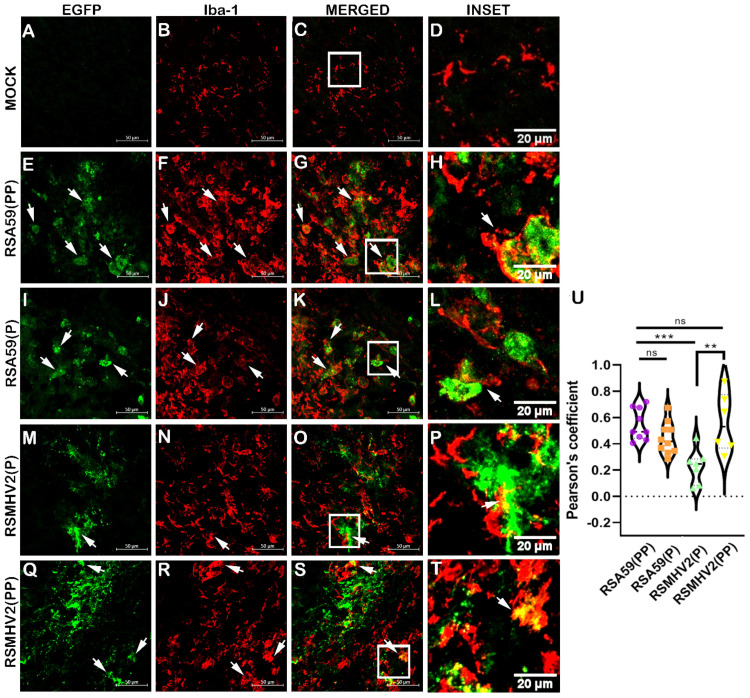
Colocalization study of two proline containing strains RSA59(PP), RSMHV2(PP) and single proline containing strains RSMHV2(P), RSA59(P) infected cells with microglia in the brain of mice at day 5 p.i. Cryosections of the brain of infected mice and mock-infected mice were immunofluorescently labeled with Iba-1 (**B**,**F**,**J**,**N**,**R**). EGFP marks the infected cells in panels (**E**,**I**,**M**,**Q**), and mock in panel (**A**). Green and red channels are merged (**C**,**G**,**K**,**O**,**S**) and show colocalization of EGFP with Iba-l-positive cells with arrowheads highlighting virus infected microglia. Mock infected sections are shown in panels (**A**–**D**), RSA59(PP) infected sections in panels (**E**–**H**), RSA59(P) infected sections in panels (**I**–**L**), RSMHV2(P) infected sections in panels (**M**–**P**), and RSMHV2(PP) infected sections in panels (**Q**–**T**). The insets marked by a white rectangle in the merged panels show a magnified view of the colocalization of microglia with EGFP (seen as yellow) in panels (**D**,**H**,**L**,**P**,**T**). Quantification of infected microglia as the Pearson coefficient was done and plotted in a violin diagram (**U**). The One-way ANOVA was used to compare RSA59(PP) with RSA59(P), RSMHV2(P), RSMHV2(PP), and RSMHV2(P) with RSMHV2(PP). The level of significance was taken at. ** *p* < 0.001, *** *p* < 0.0001, and ns denotes non-significance.

**Figure 5 viruses-15-00215-f005:**
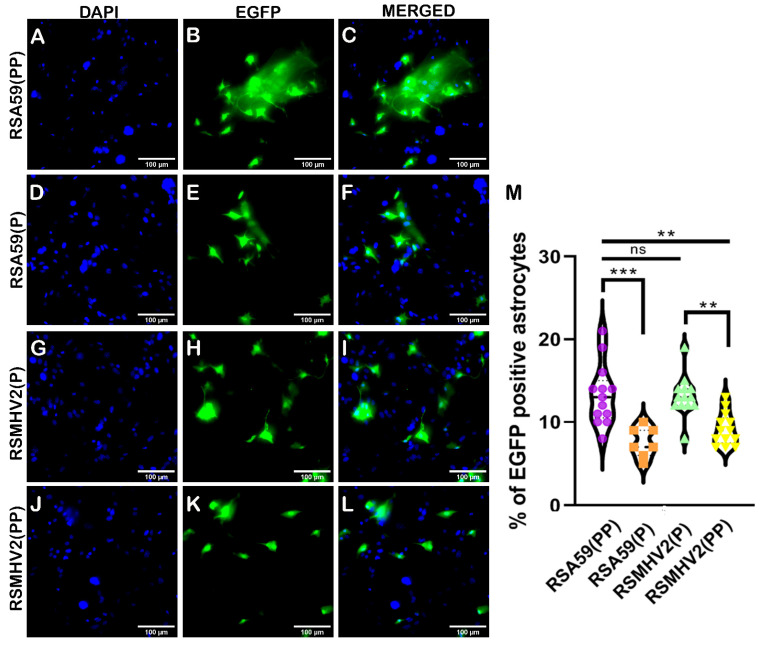
Infection of primary astrocyte cultures with parental strains RSA59(PP), RSMHV2(P), one proline deleted RSA59(P), and one proline added RSMHV2(PP) mutants. Primary astrocyte cultures were infected with RSA59(PP) (**A**–**C**), RSA59(P) (**D**–**F**), RSMHV2(P) (**G**–**I**), and RSMHV2(PP) (**J**–**L**) at MOI 2 and counterstained with DAPI at 24 h p.i. Merged images of EGFP and DAPI channels presented show infected cells and their nuclei (**C**,**F**,**I**,**L**). The percentage of EGFP-positive cells in RSA59(PP), RSA59(P), RSMHV2(P), and RSMHV2(PP) infected cultures were quantified and plotted in a violin diagram (**M**). Experiments were performed three times in triplicate per strain (N = 9). The level of significance was taken at *p* < 0.05 following One-way ANOVA analysis. ** *p* < 0.001, *** *p* < 0.0001, and ns denotes non-significance.

**Figure 6 viruses-15-00215-f006:**
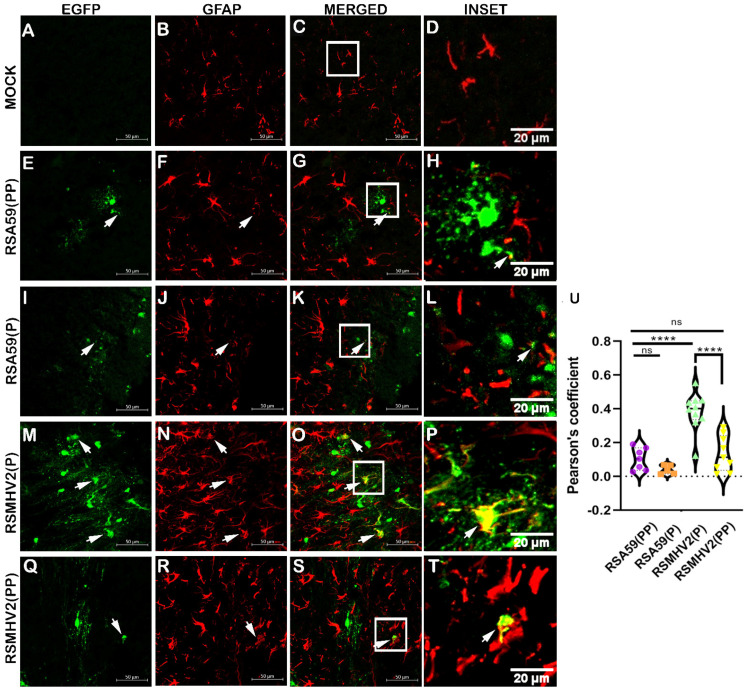
Colocalization of parental strains RSA59(PP), RSMHV2(P), one proline deleted RSA59(P), and one proline added RSMHV2(PP) mutants infected cells within astrocytes in the brain of mice at day 5 p.i. Cryosections of infected mouse brains and mock-infected mice were immunofluorescently labeled with GFAP (**B**,**F**,**J**,**N**,**R**). EGFP denotes infected cells in Panel (**E**,**I**,**MQ**) and mock-infected brain in panel A. Representative merged images of EGFP and GFAP channels (**C**,**G**,**K**,**O**,**S**) show colocalization of EGFP and GFAP-positive cells with arrowheads highlighting virus infected astrocytes. Mock sections are in Panel (**A**–**D**), RSA59(PP) infected sections are in panel (**E**–**H**), RSA59(P) infected sections are in Panel (**I**–**L**), RSMHV2(P) infected sections are in Panel (**M**–**P**), and RSMHV2(PP) infected sections are in Panel (**Q**–**T**). The insets marked by a white rectangle area in the merged panels show an area of magnified view of the colocalization of the astrocyte with EGFP (seen as yellow). Quantification of colocalization of astrocytes with the virus as the Pearson coefficient was done and plotted in a violin diagram (**U**). The One-way ANOVA is used to compare RSA59(PP) with RSA59(P), RSMHV2(P), RSMHV2(PP) and RSMHV2(P) with RSMHV2(PP). The level of significance was taken at, *p* < 0.05, **** *p* < 0.00001, and ns denotes non-significance.

**Figure 7 viruses-15-00215-f007:**
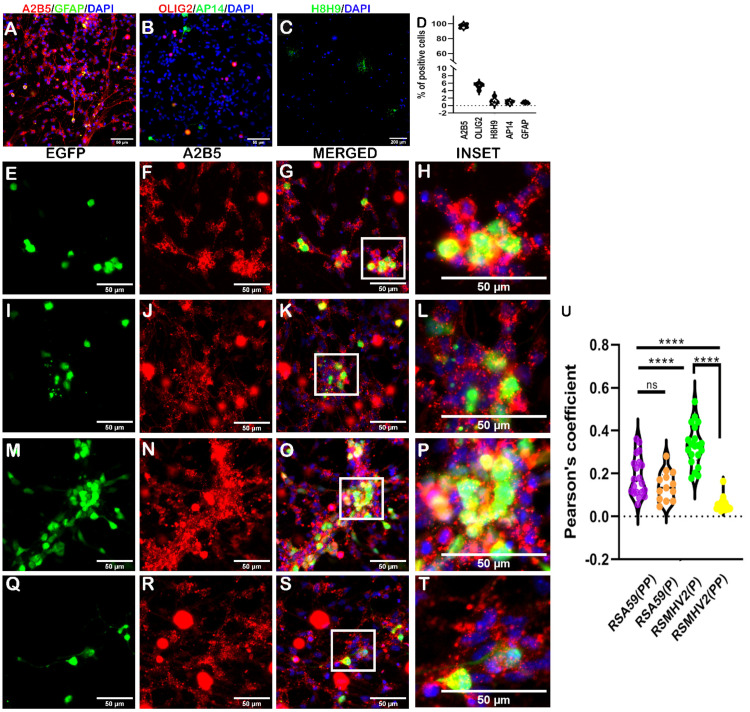
Characterization and infection of oligodendrocyte precursor enriched primary cultures with two proline containing strains RSA59 (PP), and RSMHV2(PP), and single proline containing strains RSA59(P), and RSMHV2(P). Oligodendrocyte precursor cells (OPCs) enriched from mixed glial cultures were immunolabeled with anti-A2B5 (Red; OPCs marker) and anti-GFAP (Green; astrocyte marker) (**A**), anti-Olig2 (Red; mature oligodendrocyte) and anti-AP14 (Green; mature neuron) (**B**), and anti-H8H9 (differentiated oligodendrocyte) (**C**). Cells were DAPI counterstained (Blue; nuclear stain). Visual counting and manual analysis of immunostained cells demonstrated that cultures were mostly A2B5-positive at 98%, with few cells positive for mature oligodendrocytes (**D**). OPCs cultures were infected with RSA59(PP) (**E**–**G**), RSA59(P) (**I**–**L**), RSMHV2(P) (**M**–**P**), and RSMHV2(PP) (**Q**–**T**) at MOI 2. Cells were immunolabeled with A2B5 at 24 h p.i. and counterstained with DAPI. Images presented show infected cells in the EGFP panels and A2B5 panels denote OPCs in the culture. Merged panels show the OPCs and EGFP colocalization. Panels (**H**,**L**,**P**,**T**) show higher magnification of colocalization in insets highlighted by a white rectangle area in merged images. A quantified percentage of infected OPCs was presented in a violin diagram (**U**). Experiments were performed three times in triplicate per strain (N = 9). The level of significance was taken at *p* < 0.05 following One-way ANOVA analysis. **** *p* < 0.00001, and ns denotes non-significance.

**Figure 8 viruses-15-00215-f008:**
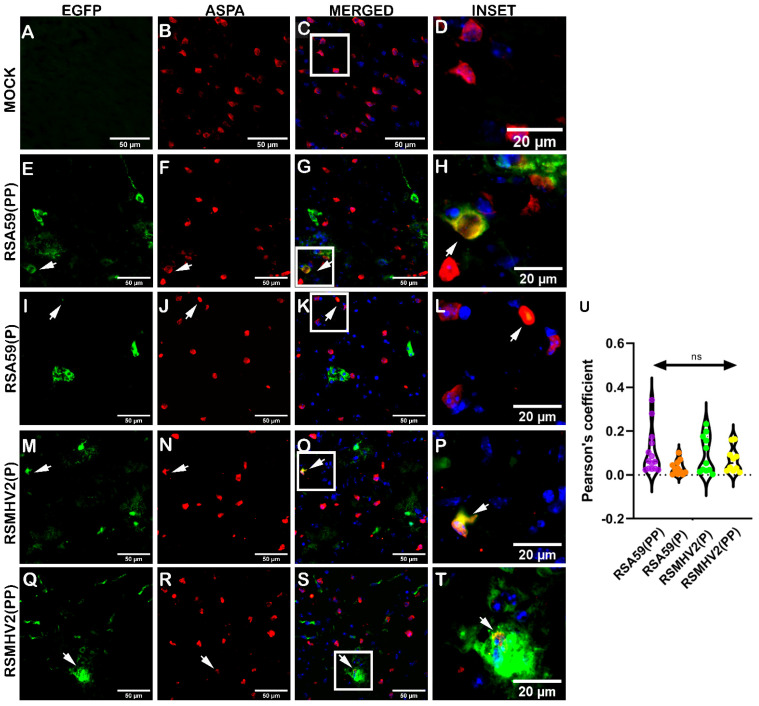
Colocalization of two proline containing strains RSA59(PP), RSMHV2(PP), single proline containing strains RSA59(P) and RSMHV2(P), infection within oligodendrocytes in the brain of mice at day 5 p.i. Cryosections of infected mouse brains and mock-infected mice were immunofluorescently labeled with ASPA (**B**,**F**,**J**,**N**,**R**). EGFP indicated infected cells in panels (**E**,**I**,**M**,**Q**), and a mock infected brain is shown in panel (**A**). Images of EGFP and ASPA were merged and DAPI counterstained (**C**,**G**,**K**,**O**,**S**) and show colocalization of EGFP and ASPA-positive cells with arrowheads highlighting virus infected oligodendrocytes. Infected brain sections from RSA59(PP), RSA59(P), RSMHV2(P), and RSMHV2(PP) are shown in Panels (**E**–**H**,**I**–**L**,**M**–**P**,**Q**–**T**), respectively. The insets highlighted by a white rectangle in the merged panels shows the area magnified view to show colocalization of oligodendrocytes with EGFP (**D**,**H**,**L**,**P**,**T**). The percentage of infected oligodendrocytes was determined and plotted in a violin diagram (**U**). The One-way ANOVA was used to compare RSA59(PP) with RSA59(P), RSMHV2(P), RSMHV2(PP) and RSMHV2(P) with RSMHV2(PP). The level of significance was taken at *p* < 0.05. ns denotes non-significance.

**Figure 9 viruses-15-00215-f009:**
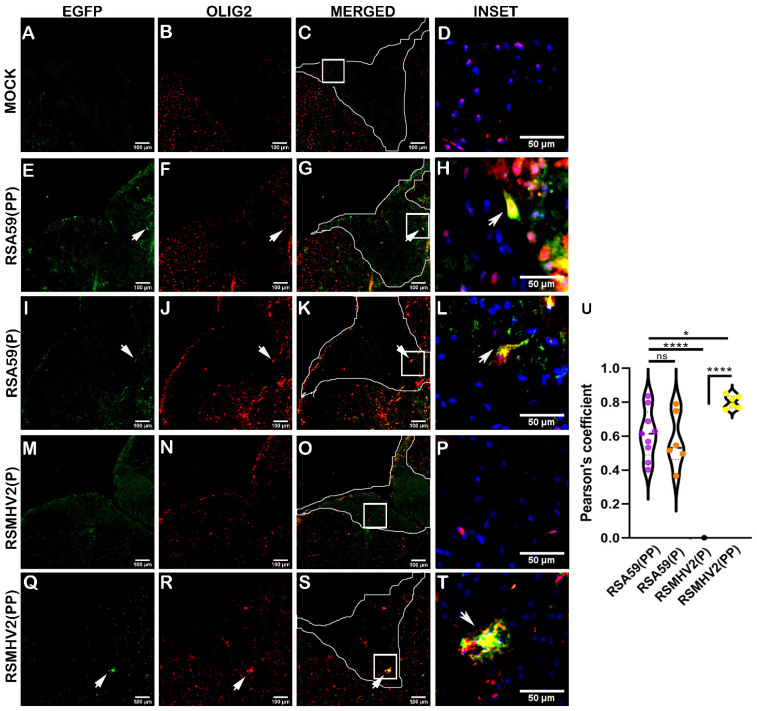
Colocalization of two proline containing strains RSA59(PP), RSMHV2(PP), and single proline containing strains RSA59(P), and RSMHV2(P) infection within mature oligodendrocytes in mouse spinal cords at day 5 p.i. Cryosections of infected mouse spinal cords alongside mock-infected mice were immunofluorescently labeled with Olig2 (**B**,**F**,**J**,**N**,**R**). EGFP denotes viral antigens in infected mice (**E**,**I**,**M**,**Q**) and mock-infected mice (**A**). Images of EGFP and Olig2 were merged with an outline of dorsal white matter highlighted in Panel (**C**,**G**,**K**,**O**,**S**). Arrowheads indicate virus infected oligodendrocytes. Higher magnification of colocalization of EGFP and Olig2-positive cells in the white matter is indicated as insets highlighted by a white rectangle area and are presented in panel (**D**,**H**,**L**,**P**,**T**). RSA59(PP) infected sections are shown in Panel (**E**–**H**), RSA59(P) infected sections are shown in Panel (**I**–**L**), RSMHV2(P) infected sections are shown in Panel (**M**–**P**), and RSMHV2(PP) infected sections are shown in Panel (**Q**–**T**). The percentage of infected oligodendrocytes in white matter was manually calculated and plotted in a violin diagram (**U**). The One-way ANOVA was used to compare RSA59(PP) with RSA59(P), RSMHV2(P), RSMHV2(PP) and RSMHV2(P) with RSMHV2(PP). The level of significance was taken at *p* < 0.05. * *p* < 0.05, **** *p* < 0.00001, and ns denotes non-significance.

**Figure 10 viruses-15-00215-f010:**
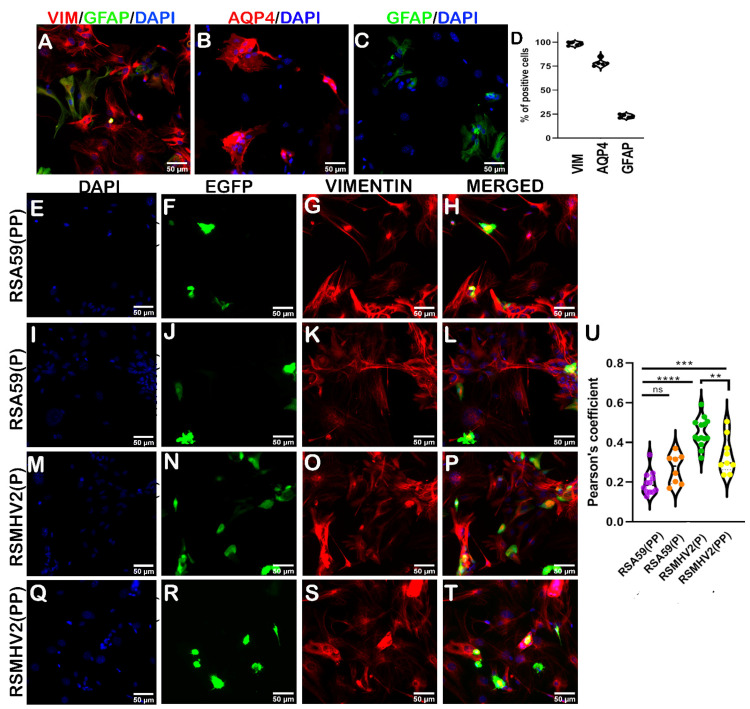
Characterization and infection of primary meningeal fibroblast cell cultures with two proline containing strains RSA59(PP), RSMHV2(PP), and single proline containing strains RSA59(P), and RSMHV2(P). Primary meningeal fibroblast cells enriched from meninges harvested from day 0–1 C57BL/6 pups were immunofluorescently characterized by double labeling with anti-vimentin (red; fibroblast marker) and anti-GFAP (green; astrocyte marker) (**A**), anti-aquaporin4 (red; matured fibroblast) (**B**), and anti-GFAP (green; astrocyte) (**C**). Cells were counterstained with DAPI. The data of the percentage of cells stained positive for the three markers were plotted (**D**). Primary meningeal fibroblast cell cultures were infected with RSA59(PP) (**E**–**H**), RSA59(P) (**I**–**L**), RSMHV2(P) (**M**–**P**), and RSMHV2(PP) (**Q**–**T**) at MOI 2. After 24 h p.i., cells were fixed and immunofluorescently labeled with vimentin and counterstained with DAPI. Representative images show infected cells in the EGFP panels (**F**,**J**,**N**,**R**) and vimentin-positive cells in the panels (**G**,**K**,**O**,**S**). Merged panels show colocalized EGFP and vimentin-stained cells. Quantification of the percentage of double-positive EGFP and vimentin cells is presented in a violin diagram (**U**). Experiments were performed three times in triplicate per strain (N = 9). The level of significance was taken at *p* < 0.05 following One-way ANOVA analysis. ** *p* < 0.001, *** *p* < 0.0001, **** *p* < 0.00001, and ns denotes non-significance.

**Figure 11 viruses-15-00215-f011:**
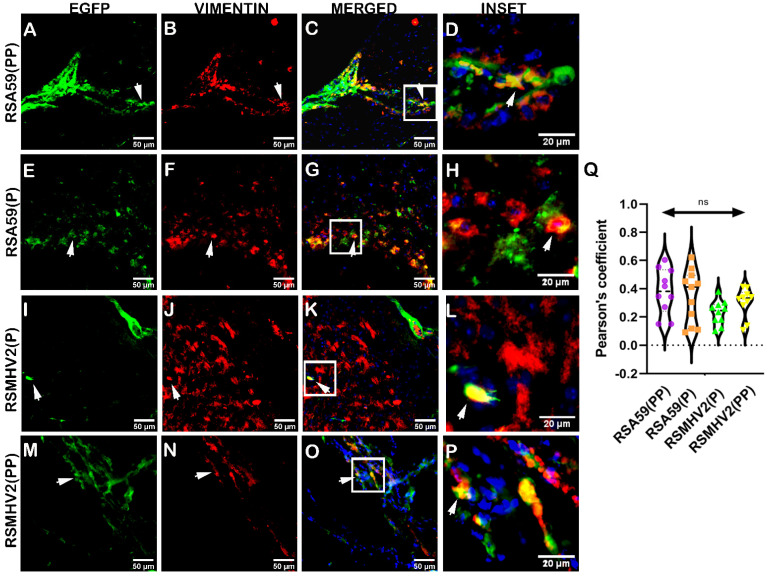
Colocalization of two proline containing strains RSA59(PP), RSMHV2(PP) and single proline containing strains RSMHV2(P) and RSA59(P) infection with choroid plexus fibroblasts in the brain of mice at day 5 p.i. Cryosections of infected mouse brains were immunofluorescently labeled with vimentin (**B**,**F**,**J**,**N**). EGFP images denote infected cells in the tissue (**A**,**E**,**I**,**M**). Images in panels (**C**,**G**,**K**, **O**) show the merged EGFP and vimentin, with insets marked by a white rectangle area and arrowheads highlighting virus infected fibroblasts. RSA59(PP) infected sections are shown in panels (**A**–**D**), RSA59(P) infected sections are shown in panels (**E**–**H**), RSMHV2(P) infected sections are shown in panels (**I**–**L**), and RSMHV2(PP) infected sections are shown in panels (**M**–**P**). A magnified view of the insets in the merged panels showed the colocalization of EGFP with vimentin-positive cells (seen as yellow) in panels (**D**,**H**,**L**,**P**). The degree of colocalization as the Pearson coefficient was plotted in the violin diagram (**Q**). The One-way ANOVA was used to compare RSA59(PP) with RSA59(P), RSMHV2(P), RSMHV2(PP) and RSMHV2(P) with RSMHV2(PP). The level of significance was taken at *p* < 0.05. ns denotes non-significance.

**Figure 12 viruses-15-00215-f012:**
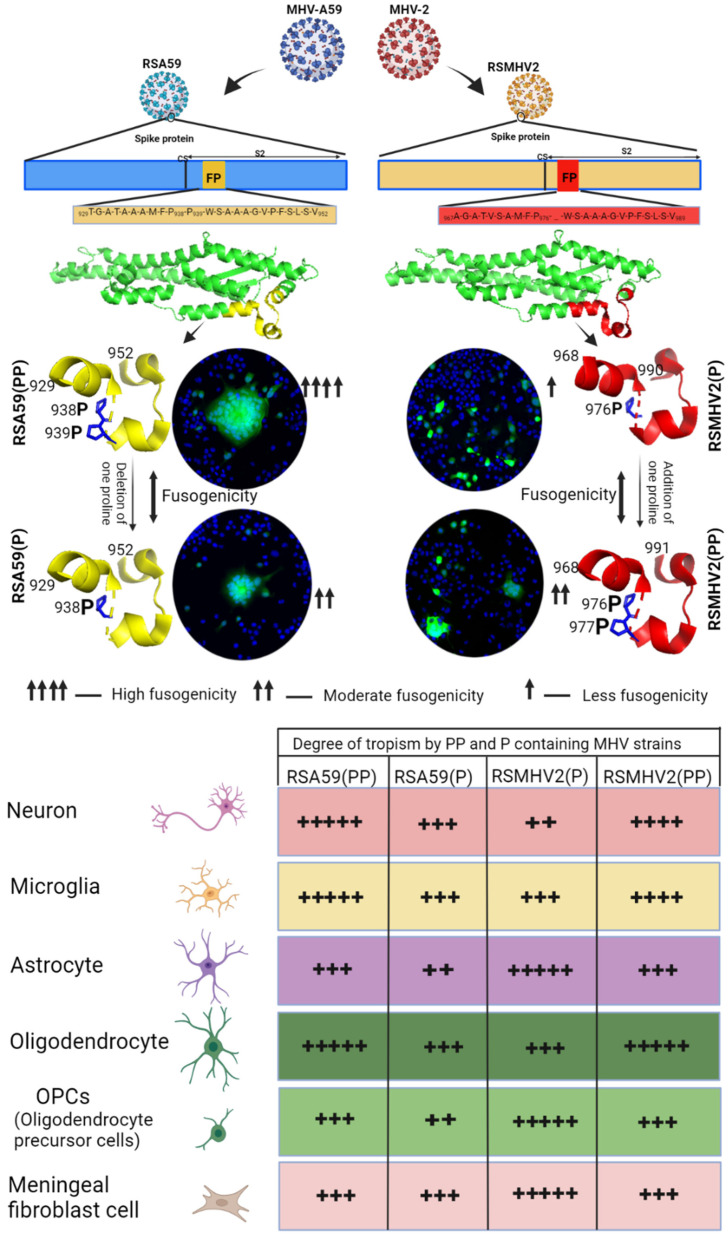
The monomeric model of the S2 subunit of Spike protein of MHV-A59, MHV-2 with their fusion peptides, the mutation and their fusogenicity and neuro-glia tropism. The cartoon diagram showed that RSA59(PP) and RSMHV2(P) were recombinant viruses engineered from MHV-A59 and MHV-2 respectively. Monomeric model of S2 subunit of MHV-A59/RSA59(PP) amino acid residues (871-1116) from PDB structure 3JCL and the S2 subunit of MHV-2 built from 3JCL for MHV-2/RSMHV2(P) amino acid residues (911-1153). The fusion peptide region highlighted in yellow and red for RSA59(PP) and RSMHV2(P) respectively, with the central proline shown in stick model. The deletion of one proline from RSA59(PP) and addition of one proline to RSMHV2(P) resulted into RSA59(P) and RSMHV2(PP), respectively. The fusogenic ability of the four viruses in N2A (neuroblastoma) cell line is indicated by up arrows with strength given in legend. The differential degree of neuroglia tropism by the four viruses is presented in table denoted by lowest degree of tropism by + and highest degree of tropism by +++++.

**Table 1 viruses-15-00215-t001:** Antibodies and respective dilutions used for immunofluorescence.

**Primary Antibody**	**Dilution**	**Secondary Antibody**	**1:800**
Rabbit polyclonal Anti-MAP2 (Sigma, Saint Louis, MO, USA)	1:200	Alexa fluor 568 Donkey Anti Rabbit (Invitrogen, Waltham, MA, USA)	1:800
Rabbit polyclonal Anti-IBA1 (Wako)	1:200	Alexa fluor 568 Donkey Anti Rabbit (Invitrogen, Waltham, MA, USA)	1:800
Mouse monoclonal Anti-Glial Fibrillary Acidic Protein antibody (GFAP) (Sigma, Saint Louis, MO, USA)	1:200	TRITC Goat Anti Mouse IgG (Jackson Immunoresearch, West Grove, PA, USA)	1:250
Rabbit polyclonal Anti-Olig2 (Sigma, Saint Louis, MO, USA)	1:100	Alexa fluor 596 Tyramide Goat Anti Rabbit IgG (Invitrogen, Waltham, MA, USA)	
Rabbit polyclonal Anti-Aspartoacylase antibody (ASPA) (Invitrogen, Waltham, MA, USA)	1:200	Alexa fluor 568 Goat Anti Rabbit (Invitrogen, Waltham, MA, USA)	1:1000
Rabbit polyclonal Anti-Vimentin (Invitrogen, Waltham, MA, USA)	1:200	Alexa fluor 568 Goat Anti Rabbit (Invitrogen, Waltham, MA, USA)	1:1000
Rabbit polyclonal Anti-Aquaporin4 (Sigma, Saint Louis, MO, USA)	1:200	Alexa fluor 568 Goat Anti Rabbit (Invitrogen, Waltham, MA, USA)	1:1000
Anti-A2B5 (mouse monoclonal antibody against precursors of oligodendrocytes)	1:50	TRITC Goat Anti Mouse IgG (Jackson Immunoresearch, West Grove, PA, USA)	1:250
Anti-AP14 (mouse monoclonal antibody against mature neurons) Anti-H8H9 (same as anti-Gal C) (mouse monoclonal antibody against mature Oligodendrocytes)	1:10 1:50	FITC Goat Anti Mouse IgG (Jackson Immunoresearch, West Grove, PA, USA) FITC Goat Anti Mouse IgG (Jackson Immunoresearch, West Grove, PA, USA)	1:250 1:250

## Data Availability

Not applicable.

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
