# Peer review of "Proline–Proline Dyad in the Fusion Peptide of the Murine β–Coronavirus Spike Protein’s S2 Domain Modulates Its Neuroglial Tropism"

_viruses, 2023, doi:10.3390/v15010215_

Round 1

Reviewer 1 Report

This study evaluated the tropism of MHV containing one or 2 prolines in the fusogenic region of the spike protein. The authors calculated the number of virus infected neuronal cells in vitro and in vivo from microscopic images. I was not totally convinced by the microscopy (see below) and the manuscript had a number of grammatical issues. Some of these are described below.

 1. Line 12. two-consecutive-central-prolines

I wasn’t sure that these words should be connected

2.Line 15. toits

3. Line 16. These studies highlighted the crucial role of PP in fusogenicity,

Insert the word “studies”

4. Line 17. Combining computational studies with biophysical data indicate that PP at the center of the FP provides local rigidity while imparting global fluctuation to the Spike proteinthat enhances the fusogenic properties of RSA59(PP) and RSMHV2(PP).

Would read better “Computational studies, combined with biophysical data indicates that PP at the center of the FP provides local rigidity while imparting global fluctuation to the Spike protein that enhances the fusogenic properties of RSA59(PP) and RSMHV2(PP). 19

5. Line 22. PP significantly enhances the tropism of neurons, microglia, and oligodendrocytes. PP is, however not essential for either astroglial or oligodendroglial precursors’ tropism, or the infection of meningeal fibroblast in the blood-brain and blood-CSF barrier.

As written this sentence implies that neuronal cells have tropism for the virus rather than the other way round. Would read better “PP significantly enhances viral tropism for neurons, microglia, and oligodendrocytes. PP is, however is not essential for viral tropism for either astroglial or oligodendroglial precursors’, or the infection of meningeal fibroblasts in the blood-brain, and blood-CSF, barrier.

6. The coronaviridae family is the largest known enveloped, non-segmented posi- 33 tive–sense polarity RNA viruses

Would read better “The coronaviridae family is the largest known group of enveloped, non-segmented positive–sense polarity RNA viruses

7. Line 35. The disease outcomes could be diverse and related to the respiratory, enteric, hepatic, and central 36 nervous systems [1, 2].

Would read better “Disease out comes are diverse and can affect the respiratory, enteric, hepatic, and central 36 nervous systems [1, 2].

8. Line 37. Would read better “The β-coronavirus genus, poses a  particular threat to humanity as indicated by recent outbreaks of human-CoV, including SARS-CoV-2 which caused the COVID-19 pandemic, as well as SARS-CoV and MERS-CoV which caused limited, but lethal, outbreaks.

9. Line 344 ……..showed that the four variants of MHV infect the neurons effectively with differential efficiency (Fig.1).

I might just lose the word effectively here

10. Line 349. As previously reported[38], fusogenicity and axonal spread are significantly observed in RSA59(PP) and 350 RSMHV2(PP). In contrast, RSA59(P) revealed a much less axonal spread of viral antigen. RSMHV2(P) showed single-cell infection of majorly non-neuronal cells in the culture as shown by a large number of EGFP with less colocalization with MAP2 positive cells (Fig. 353 1I-L).

How exactly did the authors determine axonal spread of virus?

11. Colocalization of virus in neuronal cells was a bit difficults to see in some of the images – neurons and with astrocytes. In astrocytes it was particularly hard. Perhaps the authors could put arrows to show infected cells.

Author Response

This study evaluated the tropism of MHV containing one or 2 prolines in the fusogenic region of the spike protein. The authors calculated the number of virus infected neuronal cells in vitro and in vivo from microscopic images. I was not totally convinced by the microscopy (see below) and the manuscript had a number of grammatical issues. Some of these are described below.

  1. Line 12. two-consecutive-central-prolines

I wasn’t sure that these words should be connected

2.Line 15. toits

  1. Line 16. These studieshighlighted the crucial role of PP in fusogenicity,

Insert the word “studies”

  1. Line 17. Combining computational studieswith biophysical data indicate that PP at the center of the FP provides local rigidity while imparting global fluctuation to the Spike proteinthat enhances the fusogenic properties of RSA59(PP) and RSMHV2(PP).

Would read better “Computational studies, combined with biophysical data indicates that PP at the center of the FP provides local rigidity while imparting global fluctuation to the Spike protein that enhances the fusogenic properties of RSA59(PP) and RSMHV2(PP). 19

  1. Line 22. PP significantly enhances the tropism of neurons, microglia, and oligodendrocytes. PP is, however not essential for either astroglial or oligodendroglial precursors’ tropism, or the infection of meningeal fibroblast in the blood-brain and blood-CSF barrier.

As written this sentence implies that neuronal cells have tropism for the virus rather than the other way round. Would read better “PP significantly enhances viral tropism for neurons, microglia, and oligodendrocytes. PP is, however is not essential for viral tropism for either astroglial or oligodendroglial precursors’, or the infection of meningeal fibroblasts in the blood-brain, and blood-CSF, barrier.

  1. The coronaviridae family is the largest known enveloped, non-segmented posi- 33 tive–sense polarity RNA viruses

Would read better “The coronaviridae family is the largest known group of enveloped, non-segmented positive–sense polarity RNA viruses

  1. Line 35. The disease outcomes could be diverse and related to the respiratory, enteric, hepatic, and central 36 nervous systems [1, 2].

Would read better “Disease out comes are diverse and can affect the respiratory, enteric, hepatic, and central 36 nervous systems [1, 2].

  1. Line 37. Would read better “The β-coronavirus genus, poses a  particular threat to humanity as indicated by recent outbreaks of human-CoV, including SARS-CoV-2 which caused the COVID-19 pandemic,as well as SARS-CoV and MERS-CoV which caused limited, but lethal, outbreaks.

  1. Line 344 ……..showed that the four variants of MHV infect the neurons effectively with differential efficiency (Fig.1).

I might just lose the word effectively here

  1. Line 349. As previously reported[38], fusogenicity and axonal spread are significantly observed in RSA59(PP) and 350 RSMHV2(PP). In contrast, RSA59(P) revealed a much less axonal spread of viral antigen. RSMHV2(P) showed single-cell infection of majorly non-neuronal cells in the culture as shown by a large number of EGFP with less colocalization with MAP2 positive cells (Fig. 353 1I-L).

How exactly did the authors determine axonal spread of virus?

We agree with the reviewer that we might not be able to detect axonal cells without a high-resolution microscope which is why we have changed the term to the neuronal spread of the virus.

  1. Colocalization of virus in neuronal cells was a bit difficults to see in some of the images – neurons and with astrocytes. In astrocytes it was particularly hard. Perhaps the authors could put arrows to show infected cells.

We do agree with the reviewer, boxes are difficult to interpret we will add some arrowheads to emphasize the colocalization points.   

Reviewer 2 Report

Safiriyu et al. described the analysis of the neurological cell tropism of the recombinant Mouse Hepatitis Virus with the difference in their FP region in in vitro and in vivo. Specifically, they tested one-proline or two-proline strains in the central region of FP in the background of (i) fusogenic RSA59 whose S is derived from neurotropic MHV-A59 and (ii) non-fusogenic RSMHV2 strains which has S from hepatotropic MHV-2. They challenged in vitro cultured cells enriched in a particular cell type of interest such as neurons, glia cells and so on with the recombinant viruses. A corresponding similar infection study was done in vivo using mice.

They used colocalization of signals derived from virus-expressed GFP and particular cell surface markers as the evidence of the infection of the particular cell types. 

General comments:

Though their in vitro cell cultures are not 100% pure, their claims that PP may play a role on the tropism for nurons, microglia, and oligo dendrocytes, but not for astroglia, oligo dendrocyte precursors, and fibroblasts were supported by their data. The authors should address the following issues to make their study more appealing.

The spacing errors between words (no space between two words) are scattered all over the manuscript and it severely reduced the readability. They should proof read the manuscript carefully. The tense of sentences is also confusing; the “Materials and Methods” and “Results” sections should use the past not present tense to describe what they have done or observed. They are again disturbing to read. 

Specific comments.

1)    They claim PP is crucial for fusogenicity, but it seems it is cell type dependent as shown in Fig.3 and Fig.5 where there seemed no difference in RSMHV2 background. To this reviewer, it is rather affected by the background of S; i.e. S from MHV-A59 is more fusogenic with PP. Is the fusogenicity really a factor for tropism like authors claim? What cell types was the original fusogenicity evaluated?

2)    The fusogenicity is affected by the amount of S protein available on cell surface as well as its receptors. What is the expression level of the S protein or receptors in each cell types?

3)    Related to the point#2, what is the quality (e.g. proteolysis efficacy) and amount of S in the cell or on cell surface?

4)    In in vivo study, what is the preference of the infection (or kinetics of infection) in each different cell types? Are the syncytia observed really derived from the infection of one particular cell type not from the secondary fusion between the different cell types? If authors performed the infection studies in mixed cell cultures, which cells will be infected first?

5)    Is PP motif present in SARSCoV2 and other coronaviruses as authors claimed that the FP-targeted therapy may be relevant?

Author Response

Comments and Suggestions for Authors

Safiriyu et al. described the analysis of the neurological cell tropism of the recombinant Mouse Hepatitis Virus with the difference in their FP region in in vitro and in vivo. Specifically, they tested one-proline or two-proline strains in the central region of FP in the background of (i) fusogenic RSA59 whose S is derived from neurotropic MHV-A59 and (ii) non-fusogenic RSMHV2 strains which has S from hepatotropic MHV-2. They challenged in vitro cultured cells enriched in a particular cell type of interest such as neurons, glia cells and so on with the recombinant viruses. A corresponding similar infection study was done in vivo using mice.

They used colocalization of signals derived from virus-expressed GFP and particular cell surface markers as the evidence of the infection of the particular cell types. 

General comments:

Though their in vitro cell cultures are not 100% pure, their claims that PP may play a role on the tropism for nurons, microglia, and oligo dendrocytes, but not for astroglia, oligo dendrocyte precursors, and fibroblasts were supported by their data. The authors should address the following issues to make their study more appealing.

The spacing errors between words (no space between two words) are scattered all over the manuscript and it severely reduced the readability. They should proof read the manuscript carefully. The tense of sentences is also confusing; the “Materials and Methods” and “Results” sections should use the past not present tense to describe what they have done or observed. They are again disturbing to read. 

Specific comments.

1)    They claim PP is crucial for fusogenicity, but it seems it is cell type dependent as shown in Fig.3 and Fig.5 where there seemed no difference in RSMHV2 background. To this reviewer, it is rather affected by the background of S; i.e. S from MHV-A59 is more fusogenic with PP. Is the fusogenicity really a factor for tropism like authors claim? What cell types was the original fusogenicity evaluated?

We thank the reviewer for their comment. In our previous studies, the fusogenicity induced by RSA59 (MHVA59(PP)) was studied by infecting L2 cells, rat lung epithelial cells where profuse syncytia were observed [1, 2]. Another study showed profuse syncytia formation in oligodendrocyte precursor and differentiated oligodendrocytes upon RSA59 infection in vitro in their enriched primary culture ([3]Fig. 3 and 4).

2)    The fusogenicity is affected by the amount of S protein available on cell surface as well as its receptors. What is the expression level of the S protein or receptors in each cell types?

We did not check the level of S protein in different cell types due to a lack of antibodies in our lab and this is beyond the scope of this manuscript.

3)    Related to the point#2, what is the quality (e.g. proteolysis efficacy) and amount of S in the cell or on cell surface?

As mentioned in the previous response, we have not been able to check the quality of the S protein in the cell or on its surface and this is beyond the scope of this manuscript.

4)    In in vivo study, what is the preference of the infection (or kinetics of infection) in each different cell types? Are the syncytia observed really derived from the infection of one particular cell type not from the secondary fusion between the different cell types? If authors performed the infection studies in mixed cell cultures, which cells will be infected first?

The previously unpublished data indicates that oligodendrocytes are the first to get infected by the virus followed by infection of astrocytes along with meningeal fibroblast cells and Neuronal cells are the last to get infected (Data not shown). The syncytia observed are a result of infection of both of one particular cell type and in mixed glial culture or in vivo in secondary fusion between different cell types ([3]Fig. 7).  

5)    Is PP motif present in SARSCoV2 and other coronaviruses as authors claimed that the FP-targeted therapy may be relevant?

Yes, PP is present in the fusion peptide of SARS-CoV2 [4].

The SARS-CoV-2 fusion peptides have unique physicochemical properties, accrued in part from the presence of consecutive prolines that impart backbone rigidity which aids the virus fusogenicity. The specific contribution of these prolines has been inferred from comparative studies of their deletion mutant in a fellow murine β-coronavirus MHV-A59 that shows significantly diminished fusogenicity in vitro and associated pathogenesis in vivo [2, 4]. The Spike cleavage-linked priming and fusogenic conformational transition steered by the fusion loop may be critical for the SARS-CoV-2 spread.

Previous studies highlighted the importance of the fusion loop region which could be a legitimate target for the design of vaccines or synthetic agents for therapy against COVID-19. The synergy brought about by the global location of the surface exposed fusion peptides, their physicochemical features, and the fusogenic conformational transition appears to drive the fusion process, which may explain the severity of the infection and the widespread nature of the COVID-19 pandemic [4].

  1. Safiriyu AA, Singh M, Kishore A, Mulchandani V, Maity D, Behera A, Sinha B, Pal D, Das Sarma J. 2022. Two Consecutive Prolines in the Fusion Peptide of Murine β-Coronavirus Spike Protein Predominantly Determine Fusogenicity and May Be Essential but Not Sufficient to Cause Demyelination. Viruses 14.
  2. Singh M, Kishore A, Maity D, Sunanda P, Krishnarjuna B, Vappala S, Raghothama S, Kenyon LC, Pal D, Das Sarma J. 2019. A proline insertion-deletion in the spike glycoprotein fusion peptide of mouse hepatitis virus strongly alters neuropathology. J Biol Chem 294:8064-8087.
  3. Kenyon LC, Biswas K, Shindler KS, Nabar M, Stout M, Hingley ST, Grinspan JB, Das Sarma J. 2015. Gliopathy of Demyelinating and Non-Demyelinating Strains of Mouse Hepatitis Virus. 9.
  4. Pal D. 2021. Spike protein fusion loop controls SARS-CoV-2 fusogenicity and infectivity. J Struct Biol 213:107713.

Reviewer 3 Report

Safiriyu et al. researched the relationships between neuron/neuroglial tropism and PP-dyad of fusion domain in MHV infection. Their efforts for obtaining the datas from 4 viral strains and the diverse primary culture cells and tissues were respectable. However, their manuscripts including core data had the problems for appropriate study, I think.     [Major points] 1) Entire pictures in the figures are too coarse to evaluate the infection to each neuronal cells, especially  in vivo figures. One of the problem is low resolution. Secondly, without DAPI staining, cells in the tissues are not distinguishable, therefore infected cells were not properly evaluated.   -fig1; Using the Neu-N for neuron marker are recommended. thus EGFP/NeuN/DAPI positive cells were clearly infected neurons -fig2; add DAPI staining, and using the Neu-N for neuron marker   -fig3; add Iba-1 staining -fig4; add DAPI staining   -fig5; add GFAP staining
-fig6; add DAPI staining

-fig8/-fig9; there are no explanations about  the change the markers, A2B5 and ASPA   -fig10; add DAPI staining   AND/OR another experiments for detect the infection, FACS analysis, are preferable, not essential.   2) Your manuscripts title is far from your experimental results. The presence or absence PP sychronizes only neurons, microglia, and oligodendrocytes, not of all kind of neuroglial cell linages.   3) Membrane lipid composition were noted in Table2, Is it essential for your conclusion or discussion? Not deeply mentioned in the discussion section. that was confusing. Furthermore, TableS1 showed the no big differences between Membrane lipid composition of the cells.   4) Cartoon (2D and/or 3D structure) is needed for the readers to confirm the PP-dyad in spike protein. and background of the spike proteins example a) parental RSA59 (PP), derived from MHV-A59, fusogenic + b) mutated RSA59 (P), fusogenic - c) parental RSMHV (P) derived from MHV-2, fusogenic - d) mutated RSA59 (PP), fusogenic +   5) Novelty of the data and study concepts should be re-confirmed. Several your previous work on the another cell line or neuronal cell linage were published. Emphasis on the difference of the previous works, concisely. And  introducing previous works should be more  concise.     [Minor points]
1) SO MANY typos No spaces between words were too much, check them.   2) Scale bars INSET pics and retangle area are slightly out of alignment. The font of scale bar in figures are almost too little, In Fig2, 4, 6, scale bars in INSET is wrong.   3) Text words It might be so many words in the text. concising the description is recommended.

Author Response

Comments and Suggestions for Authors

Safiriyu et al. researched the relationships between neuron/neuroglial tropism and PP-dyad of fusion domain in MHV infection. Their efforts for obtaining the datas from 4 viral strains and the diverse primary culture cells and tissues were respectable. However, their manuscripts including core data had the problems for appropriate study, I think.  

[Major points] 

1) Entire pictures in the figures are too coarse to evaluate the infection to each neuronal cells, especially  in vivo figures. One of the problem is low resolution. Secondly, without DAPI staining, cells in the tissues are not distinguishable, therefore infected cells were not properly evaluated.   -fig1;

Using the Neu-N for neuron marker are recommended. thus EGFP/NeuN/DAPI positive cells were clearly infected neurons -fig2;

add DAPI staining, and using the Neu-N for neuron marker   -fig3;

add Iba-1 staining -fig4; add DAPI staining   -fig5; add GFAP staining
-fig6; add DAPI staining -fig8/-fig9;

there are no explanations about  the change the markers, A2B5 and ASPA   -fig10;

add DAPI staining   AND/OR another experiments for detect the infection, FACS analysis, are preferable, not essential.   

We agree with the reviewer that the picture quality especially in vivo figures are of low resolution but we must admit that all the images are taken in the confocal microscope. Given the complexity of the brain cells and the inflamed tissues, keeping the integrity of the cell structure is a big challenge, therefore some of the images are compromised. We have tried our level best to acquire the images to demonstrate the infectivity pattern.

Reviewers’ concern with Fig.1., in our hand Neu-N did not work as expected in the inflamed tissues but MAP2 clearly showed the infected neurons for that reason Neu-N data is beyond the scope of this manuscript.

As per the suggestion of the reviewer, we have added the DAPI panel in Fig.2.

In Fig. 3, and Fig.4 due to the complexity of the staining in the inflamed tissues we avoided using DAPI panel to avoid bleed-through of the images. In our previous publication, pertaining to the same bleed-through issues we avoided DAPI panel [1, 2].

2) Your manuscripts title is far from your experimental results. The presence or absence PP sychronizes only neurons, microglia, and oligodendrocytes, not of all kind of neuroglial cell linages. 

As per the suggestions of the reviewers, we have revised the title to “Proline-Proline dyad in the Fusion peptide of the Murine β–Coronaviruses Spike protein’s S2 domain determines its neuroglial tropism”.

3) Membrane lipid composition were noted in Table2, Is it essential for your conclusion or discussion? Not deeply mentioned in the discussion section. that was confusing. Furthermore, TableS1 showed the no big differences between Membrane lipid composition of the cells.

As the reviewer suggested we have removed the correlation mentioned of spike with membrane lipid composition, Table2 and TableS1.    

4) Cartoon (2D and/or 3D structure) is needed for the readers to confirm the PP-dyad in spike protein. and background of the spike proteins example  a) parental RSA59 (PP), derived from MHV-A59, fusogenic +  b) mutated RSA59 (P), fusogenic -  c) parental RSMHV (P) derived from MHV-2, fusogenic -  d) mutated RSA59 (PP), fusogenic +  

As per the suggestion, we have prepared a cartoon diagram in BioRender.  (Attached in the file below)                                                                                               

 5) Novelty of the data and study concepts should be re-confirmed. Several your previous work on another cell line or neuronal cell linage were published. Emphasis on the difference of the previous works, concisely. And introducing previous works should be more concise.    

Although several of our work has been published in neuronal-glial lineages, the novelty of this study lies in the comparison of the four proline-added or deleted mutant virus stains. The entire study emphasized the role of two central consecutive prolines in the viral spread, fusogenecity, and neuroglial tropism.  

[Minor points]
1) SO MANY typos No spaces between words were too much, check them.  

We apologize for our typological mistakes. We took care of the spacing issues.

2) Scale bars INSET pics and retangle area are slightly out of alignment. The font of scale bar in figures are almost too little, In Fig2, 4, 6, scale bars in INSET is wrong.   

We have taken care of the same in the revised manuscript. The font of the scale bar has been changed. The scale bars of Fig 2,4,6 have been corrected.

3) Text words It might be so many words in the text. concising the description is recommended.

We tried to make the manuscript as concise as possible, but to ensure the clarity of the study we chose not to shorten the manuscript further.

  1. Das Sarma J, Iacono K, Gard L, Marek R, Kenyon LC, Koval M, Weiss SR. 2008. Demyelinating and nondemyelinating strains of mouse hepatitis virus differ in their neural cell tropism. J Virol 82:5519-26.
  2. Safiriyu AA, Singh M, Kishore A, Mulchandani V, Maity D, Behera A, Sinha B, Pal D, Das Sarma J. 2022. Two Consecutive Prolines in the Fusion Peptide of Murine β-Coronavirus Spike Protein Predominantly Determine Fusogenicity and May Be Essential but Not Sufficient to Cause Demyelination. Viruses 14.

Round 2

Reviewer 2 Report

The most of points raised by this reviewer except for the issues with technical difficulties were addressed by the authors. However, more careful biochemical analyses of S proteins themselves are required in future studies to support their arguments. The word “Determines” in the title seems too strong to describe their findings. To this reviewer, it is rather "modulates".

Author Response

We thank the reviewer for their thoughtful comments, suggestions, and corrections that have made the manuscript more comprehensive and informative.

The title of the manuscript has been revised as per the suggestion of the reviewer. 

Reviewer 3 Report

Authors addressed the points I commented.

Majority points were improved, however a little points have remain insufficient.

4) Cartoon (2D and/or 3D structure) is needed for the readers to confirm the PP-dyad in spike protein. and background of the spike proteins example a) parental RSA59 (PP), derived from MHV-A59, fusogenic + b) mutated RSA59 (P), fusogenic - c) parental RSMHV (P) derived from MHV-2, fusogenic - d) mutated RSA59 (PP), fusogenic +

As per the suggestion, we have prepared a cartoon diagram in BioRender. (Attached in the file below)

- Great and understandable cartoon reminding and summarizing the previous works. Combined your results (PP affects the tropism to neurons, microglia, and oligodendrocytes. PP does not to astroglial or oligodendroglial precursors) into the cartoon diagram, insertion the revised cartoon diagram into main-text as figure 12 or graphical abstract, is strongly recommended.

3) Membrane lipid composition were noted in Table2, Is it essential for your conclusion or discussion? Not deeply mentioned in the discussion section. that was confusing. Furthermore, TableS1 showed the no big differences between Membrane lipid composition of the cells.

As the reviewer suggested we have removed the correlation mentioned of spike with membrane lipid composition, Table2 and TableS1.

- OK. Delete completely table S1 from supplemental info.

Author Response

We thank the reviewer for their thoughtful comments, suggestions, and corrections that have made the manuscript more comprehensive and informative.

The revised cartoon diagram has been uploaded as the graphical abstract and table S1 removed from the supplementary section.